# Integrated externally and internally generated task predictions jointly guide cognitive control in prefrontal cortex

**Jiefeng Jiang[1]\*, Anthony D Wagner[1,2], Tobias Egner[3,4]**

[1]Department of Psychology, Stanford University, Stanford, United States; [2]Neuroscience Program, Stanford University, Stanford, United States; [3]Center for Cognitive Neuroscience, Duke University, Durham, United States; [4]Department of Psychology and Neuroscience, Duke University, Durham, United States

**Abstract** Cognitive control proactively configures information processing to suit expected task demands. Predictions of forthcoming demand can be driven by explicit external cues or be generated internally, based on past experience (cognitive history). However, it is not known whether and how the brain reconciles these two sources of information to guide control. Pairing a probabilistic task-switching paradigm with computational modeling, we found that external and internally generated predictions jointly guide task preparation, with a bias for internal predictions. Using model-based neuroimaging, we then show that the two sources of task prediction are integrated in dorsolateral prefrontal cortex, and jointly inform a representation of the likelihood of a change in task demand, encoded in frontoparietal cortex. Upon task-stimulus onset, dorsomedial prefrontal cortex encoded the need for reactive task-set adjustment. These data reveal how the human brain integrates external cues and cognitive history to prepare for an upcoming task.
DOI: https://doi.org/10.7554/eLife.39497.001

## Introduction

'Cognitive control' describes a collection of neurocognitive mechanisms that allow us to use internal goals and ongoing context to strategically bias the manner in which we process information (*Miller and Cohen, 2001*; *Egner, 2017*). For instance, depending on current goals, humans can flexibly switch or update 'task-sets' that allow them to shift between different aspects of a stimulus to which they attend and respond (*Monsell, 2003*). Here, a task-set involves the mental representation of task-relevant stimuli, responses, and their corresponding stimulus-response mappings, allowing for an appropriate action to be executed in response to a stimulus (cf. *Monsell, 2003*; *Kiesel et al., 2010*). Cognitive control thus grants the organism considerable behavioral flexibility, but it also incurs costs, in that controlled processing is slow and effortful (*Norman and Shallice, 1986*): to wit, it takes longer to perform a trial when a shift in the task-set is required than when staying with the same set — known as 'switch costs' — which is thought to reflect additional cognitive processing, including the updating of the task-set and the resolution of proactive interference (i.e., the previously activated task-set's interference with the retrieval of the currently required task-set) (*Rogers and Monsell, 1995*; *Badre and Wagner, 2006*).

A central question about cognitive control concerns its regulation — that is, how does the brain determine when and how much control should be applied (*Botvinick et al., 2001*)? The broad suggestion is that people *predict* forthcoming task demands and adjust processing accordingly (e.g., *Shenhav et al., 2013*; *Egner, 2014*; *Jiang et al., 2014*; *Abrahamse et al., 2016*; *Waskom et al., 2017*). Importantly, such expectations about task demands can be driven by *two* sources: explicit predictions provided by external cues (*Rogers and Monsell, 1995*; *Dreisbach et al., 2002*;

\*For correspondence:
jiefeng.jiang@stanford.edu

**Competing interests:** The authors declare that no competing interests exist.

*Badre and Wagner, 2006*) and internally generated, trial history-based predictions, which are typically implicit (*Dreisbach and Haider, 2006*; *Mayr, 2006*; *Bugg and Crump, 2012*; *Egner, 2014*; *Chiu and Egner, 2017*).

Previous behavioral studies showed that the two types of predictions appear to drive control simultaneously. In particular, trial history based predictions impact cognitive control even in cases where these predictions are redundant, as in the presence of 100% valid external cues for selecting the correct control strategy (e.g., *Alpay et al., 2009*; *Correa et al., 2009*; *Kemper et al., 2016*). However, it is not presently known whether and how the brain reconciles explicit external and (typically implicit) internal predictions. Here, we sought to characterize the computational and neural mechanisms that mediate the joint influence of external, cued-based, and internal, cognitive history-based (i.e., a participant's history of cognitive processing involved in performing prior trials on this task) predictions about future task demands that drive the engagement of cognitive control. Specifically, we sought to characterize how internal and external predictions jointly affect proactive task-set updating. Note that while we make the assumption that expectations regarding forthcoming task demand can mitigate the costs of switching tasks, we here do not aim to distinguish between reduced switch costs due to improved task-set reconfiguration (*Rogers and Monsell, 1995*) versus improved resolution of mnemonic interference, or 'task-set inertia' (*Allport et al., 1994*; *Badre and Wagner, 2006*). Rather, we consider both of these processes integral components of successful task-set updating (*Qiao et al., 2017*).

We combined computational modeling of behavior on a probabilistic variant of the classic cued task switching paradigm (*Dreisbach et al., 2002*) with functional magnetic resonance imaging (fMRI) in healthy humans. On each trial, participants performed one of two perceptual decision tasks on an array of moving, colored dots (*Figure 1*). Crucially, prior to the presentation of the trial's task cue and stimulus, an explicit probabilistic *pre-cue* informed participants of the likelihood that the forthcoming trial would require performing one task (color categorization) or the other (motion categorization) (*Figure 1*). This pre-cue provided an explicit cue-induced task prediction that could be used to guide preparatory task-set updating, and be contrasted with trial history-based, internally generated predictions about the forthcoming task.

While the benefit of the explicit cue was represented by its predictive value, trial history was uninformative about the probability of task switching, as the task sequence was randomized. There was thus no objective benefit to generating trial history-based predictions. However, based on prior studies, we nevertheless anticipated that participants would form (likely implicit) internal expectations for forthcoming trials based on trial-history (e.g., *Huettel et al., 2002*), and this design ensured that trial-history and cue-based predictions were independent of each other. This allowed us to quantify the influence of each type of prediction in order to adjudicate between different possible models of control guidance, including a 'rational' model that is driven purely by the informative explicit cue.

We then leveraged the concurrently acquired fMRI data to trace neural representations of task and control demand predictions. Here, the first major question is whether externally and internally driven task predictions drive behavior in parallel or are *integrated* in a joint neural representation of task prediction. Second, it has been proposed that control processes, like the switching of a task-set, are guided by predictions of control demand (*Shenhav et al., 2013*; *Egner, 2014*; *Jiang et al., 2014*; *Abrahamse et al., 2016*; *Waskom et al., 2017*). Based on this proposal and the theory of dual mechanisms of cognitive control (*Braver, 2012*), we sought to characterize neural representations of *proactive switch demand* (the likelihood of having to switch tasks), which is determined by the relationship between the predicted forthcoming task and the task that was performed on the previous trial. Finally, our protocol also allowed us to assess the neural substrates of *reactive* cognitive control (specifically, *reactive switch demand*), based on the mismatch between task predictions and actual requirements at the time of task-stimulus onset.

To preview the results, task-switching behavior was jointly driven by internally generated and cue-induced task predictions and, strikingly, the impact of the former was stronger than that of the latter. Moreover, at the time of pre-cue onset, the fMRI data revealed an integrated representation of the joint external and internal predictions in left dorsolateral prefrontal cortex (dlPFC). This prediction informed a representation of proactive switch demand in the frontoparietal control network, and, at the time of task stimulus presentation, the prediction error associated with these joint predictions (i.e., reactive switch demand) was encoded in the dorsomedial prefrontal cortex (dmPFC), including

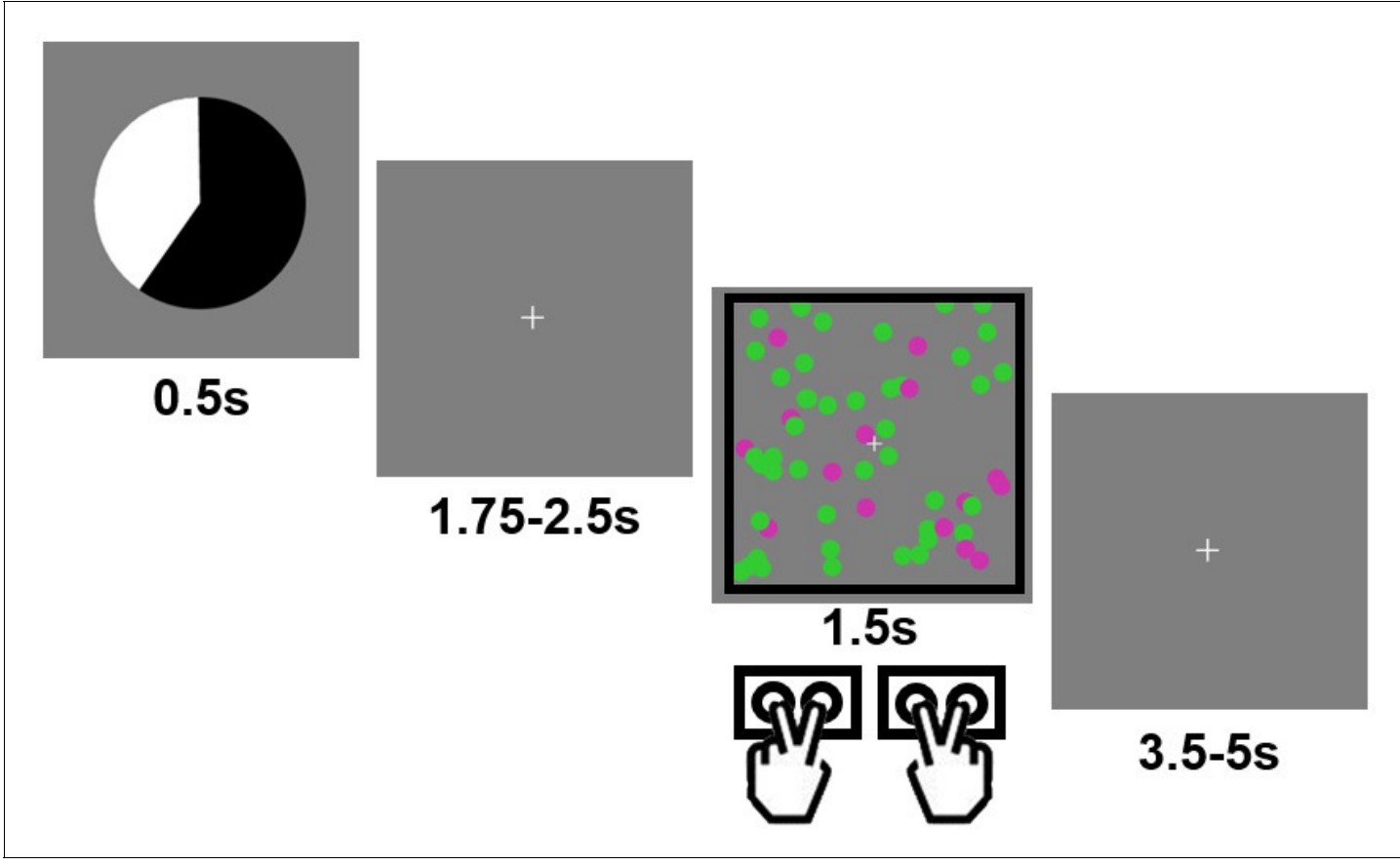

**Figure 1.** Example trial in the experimental task. Each trial started with a pie chart, whose proportions of black/white area predicted the probability of encountering a black vs. white frame surrounding the forthcoming cloud of moving, colored dots. The frame color cued the task to be performed (color vs. motion categorization).

DOI: https://doi.org/10.7554/eLife.39497.002

the anterior cingulate cortex (ACC). Collectively, these data suggest that experientially acquired and explicitly cued expectations of control demand are reconciled in dlPFC and dmPFC to jointly guide the implementation of cognitive control.

## Results

### Behavioral data – Effects of external cues and cognitive history

Participants (N = 22) performed a cued task switching protocol involving two perceptual decision-making tasks (*Figure 1*) that required reporting either the predominant color or motion direction of a noisy dot cloud stimulus. Which task to perform was indicated by the color of a simultaneously presented frame that surrounded the dot cloud. Crucially, the task stimulus was preceded by a predictive pie chart cue (i.e., the pre-cue) that accurately indicated the probability that the forthcoming task would be the color (or motion) task (five probability levels: 0.2, 0.4, 0.5, 0.6 and 0.8), and thus, whether the forthcoming trial would likely involve the same task as the previous trial (i.e., a task-repeat trial) or the other task (i.e., a task-switch trial). The task sequence itself was pseudo-random, with an equal number of switch and repeat trials occurring within each run. Participants were not told anything about the statistical properties of the task sequence, allowing us to study how subjects mix external and internal predictions without explicit knowledge of the validity of the internal prediction.

The varying benefits of cue-induced and history-based predictions in our design allowed us to adjudicate between two competing hypotheses. Based on prior behavioral literature (*Alpay et al.,*

*2009*; *Correa et al., 2009*; *Kemper et al., 2016*), cue-based and trial history-based predictions could jointly contribute to behavior (the 'joint-guidance hypothesis'). Alternatively, a rival model assumes that control strategy is driven by a 'rational' arbitration between internally generated and external predictions that is based on the expected benefit of each prediction, as represented by their respective predictive value (or certainty, cf. *Daw et al., 2005*). *Rational* is in quotations here because this model does not take into account the potential (and unknown) *costs* of employing these two types of control predictions, which may be another important factor in driving the application of control (*Kool et al., 2010*; *Shenhav et al., 2013*). Given that trial-history was not informative of the upcoming task (i.e., it had no predictive value), this alternative model in effect corresponds to control being guided exclusively by the external cue, which has predictive value. We here refer to this as the 'max-benefit hypothesis'.

These hypotheses differ in terms of how the pre-cue and trial history should drive task-set updating. Therefore, we started by analyzing how behavior was influenced by three factors: the relationship between the previous and current trial task (i.e., the task switch effect), the pre-cue probabilistic task prediction, and possible internally generated task predictions based on trial-history up to 4 trials back. We subsequently present formal modeling and model comparisons to quantify more precisely what type(s) of task prediction best accounted for behavior.

Participants performed with high accuracy (color task: 0.87 ± 0.02 [mean ±SEM]; motion task: 0.88 ± 0.02), which did not differ between tasks ($t_{21}$ = 0.30, p=0.75). To test whether the previous-trial's task (i.e., trial i-1) influenced behavior on the current trial, we conducted repeated-measures ANOVAs (previous task ×current task) on accuracy and response time (RT) data. Replicating the classic task switch cost, there was a significant interaction between previous- and current-trial tasks in both accuracy ($F_{1,21}$=47.0, p<0.001; *Figure 2A*) and RT ($F_{1,21}$=95.5, p<0.001; *Figure 2B*), driven by more accurate (task-repeat accuracy: 0.90 ± 0.01; task-switch accuracy: 0.85 ± 0.02) and faster responses (task-repeat RT: 0.87 ± 0.01 s; task-switch RT: 0.94 ± 0.02 s) when the task repeated than when it switched. Additionally, motion trials were faster than color trials ($F_{1,21}$=21.3, p<0.001; *Figure 2B*; motion RT: 0.84 ± 0.02 s; color RT: 0.97 ± 0.02 s), which is consistent with previous

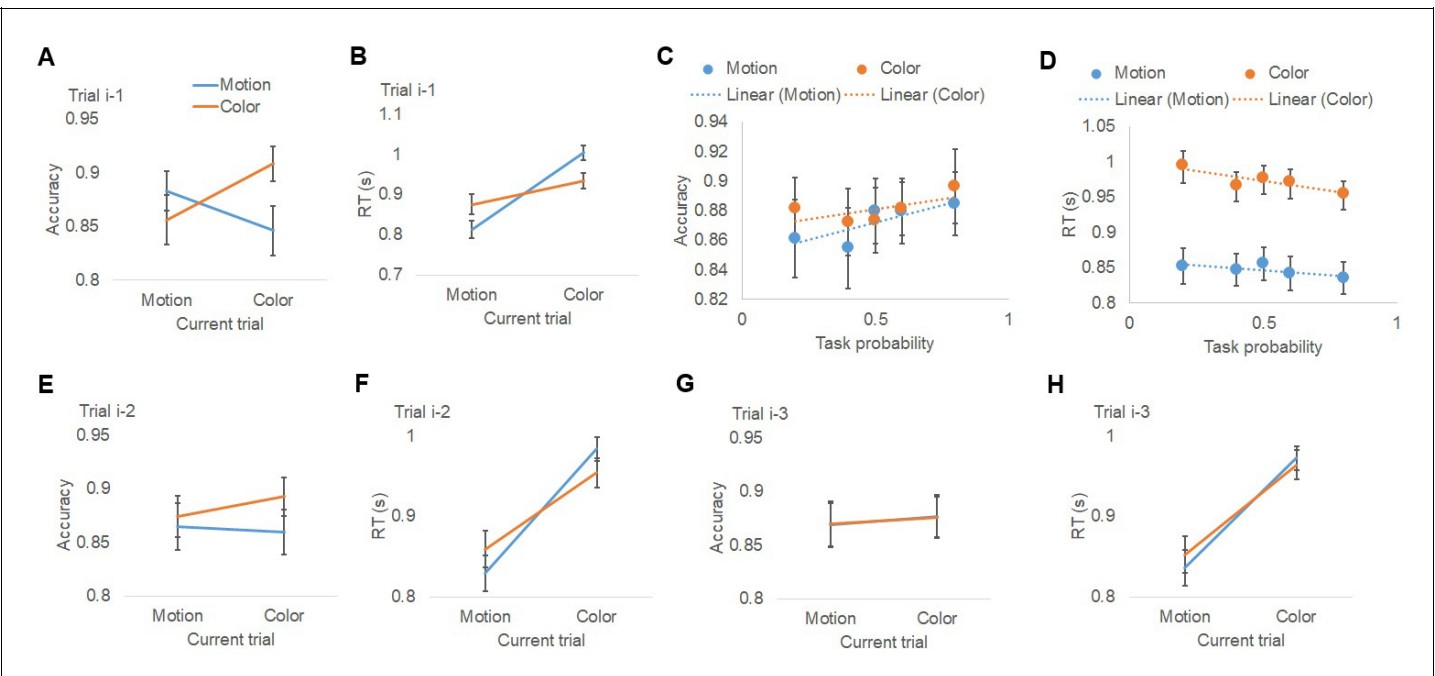

**Figure 2.** Behavioral results. (A, B) Group mean (±MSE) of accuracy (A) and RT (B), plotted as a function of task on the previous trial (i-1) and current trial. Results for trials i-2 and i-3 are plotted similarly in (E, F) and (G, H), respectively. (C, D) Group mean (±MSE) of accuracy (C) and RT (D), plotted as a function of (i) the pre-cue's prediction of encountering the actual task and (ii) the task on the current trial.
DOI: https://doi.org/10.7554/eLife.39497.003

studies using similar color/motion discrimination tasks and dot cloud stimuli (*Jiang et al., 2016*; *Waskom et al., 2017*). No other effects reached statistical significance.

To examine the effect of the probabilistic pre-cue on behavior, we performed repeated-measures ANOVAs (5 task prediction levels × current task) on accuracy and RT data (*Figure 2C,D*, see *Table 1* for summary statistics). No significant effects were detected for accuracy. Responses were again faster for motion judgments than color judgments ($F_{1,21}$=24.3, p<0.001; *Figure 2D*). Most importantly, there was a significant effect of the explicit pre-cue, as reflected by a main effect of task prediction on RT ($F_{4,84}$=4.3, p<0.005; *Figure 2D*), with response speed scaling with predictive pre-cue information. No interaction between pre-cue information and current task was observed ($F_{4,84}$ < 1), indicating that the effects of predictive cues on RT were similar in the motion and color tasks.

To assess trial-history effects beyond the immediately preceding trial, we tested whether performance was influenced by task-sets that had occurred in recent prior trials (from trials i-2 to i-4) by conducting repeated measures ANOVAs (task at prior trial × current task) on accuracy and RT. The interaction between the prior trial task-sets and the current task was significant in both accuracy ($F_{1,21}$=6.2, p=0.02; *Figure 2E*) and RT ($F_{1,21}$=73.2, p<0.001; *Figure 2F*) for trial i-2. For trial i-3, the interaction was not significant in accuracy ($F_{1,21}$<1, n.s.; *Figure 2G*), but was significant in RT ($F_{1,21}$=8.4, p=0.008; *Figure 2H*). Trial i-4 did not show a modulation effect in either accuracy or RT (both $F_{1,21}$<1, n.s). Since the modulation effect decayed as the distance from previous trial to current trial increased (*Figure 2A–B and E–H*), trials beyond i-4 were not tested. These results provide strong evidence that trial-history based task predictions — stemming from learning over the last three trials — impact behavior, biasing participants towards expecting task repetitions.

## Behavioral data – Model comparison

To formally compare how well the joint-guidance and max-benefit hypotheses explain the behavioral data, we constructed a quantitative model for each hypothesis and compared the two using trial-wise RTs. Accuracy was not modeled due to its insensitivity to external task prediction induced by the pre-cue (*Figure 2C*). We first defined three variables to represent task switching, cue-induced predictions, and internally generated predictions, respectively, with the latter two being continuous variables, ranging from 0 to 1, that represent *task-set weighting* (i.e., the relative activation of color vs. motion task-sets). Without loss of generality, a value greater than 0.5 means a prediction favoring the color task; and a value less than 0.5 indicates a prediction favoring the motion task. In particular, after the task cue was presented, the task-set became deterministic. Therefore, motion and color tasks were then represented by 0 and 1, respectively. All model variables used in this study are summarized in *Table 2*.

To capture the task switch effect, we define $T_{prev}$ as the task required on the previous trial. A task switch/repetition can thus be defined by comparing the current task to $T_{prev}$. We then denote $P_{cue}$ as the probability of encountering a color task trial based on the pre-cue. To formally model the internally generated, trial-history based task prediction, we employed a reinforcement learning model:

$$P_{int}(i) = (1 - \alpha)P_{int}(i - 1) + \alpha T_{prev} \qquad (1)$$

Where $P_{int}(i)$ encodes the internally generated prediction of the task on trial i; $\alpha$ represents the learning rate (*Figure 3A*), which is a free parameter ranging from 0 to 1 and denotes how much this prediction relies on the previous trial ($T_{prev}$) in relation to older trials (integrated in $P_{int}(i-1)$). $P_{int}(0)$ was initially set to 0.5 to reflect an unbiased initial belief of task-set. After each trial, $P_{int}$ was updated based on $T_{prev}$ and *Equation 1* and was then used as the internal prediction for the next trial.

**Table 1.** Descriptive statistics (group mean ± MSE) of behavioral data.

| Task prediction | 0.2 | 0.4 | 0.5 | 0.6 | 0.8 |
|---|---|---|---|---|---|
| Color trial accuracy | 0.88 ± 0.02 | 0.87 ± 0.02 | 0.87 ± 0.02 | 0.88 ± 0.02 | 0.90 ± 0.03 |
| Color trial RT (s) | 1.00 ± 0.02 | 0.97 ± 0.02 | 0.98 ± 0.02 | 0.97 ± 0.02 | 0.96 ± 0.02 |
| Motion trial accuracy | 0.86 ± 0.03 | 0.86 ± 0.03 | 0.88 ± 0.02 | 0.88 ± 0.02 | 0.88 ± 0.02 |
| Motion trial RT (s) | 0.85 ± 0.03 | 0.85 ± 0.02 | 0.86 ± 0.02 | 0.84 ± 0.02 | 0.84 ± 0.02 |

DOI: https://doi.org/10.7554/eLife.39497.004

**Table 2.** Summary of model variables used in this study.

Abbreviations: T = task; cur = current trial; prev = previous trial; p=prediction; int = internal.

| Variable | Meaning | Definition/range |
|---|---|---|
| $T_{cur}$ | Task required on the current trial | 0 if motion task; 1 if color task. |
| $T_{prev}$ | Task required on the previous trial | Same as above |
| $i$ | Trial index | |
| $P_{cue}$ | Probability of encountering a color task indicated by the pre-cue | 0.2, 0.4, 0.5, 0.6 or 0.8, depending on pre-cue |
| $\alpha$ | Learning rate | [0, 1] |
| $P_{int}$ | Internally generated task prediction | $(1 - \alpha)P_{int}(i-1) + \alpha T_{prev}$ |
| $P_{joint}$ | Joint task prediction | $(1-\beta)P_{int} + \beta P_{cue}$ |
| $PE_{cue}$ | Prediction error of $P_{cue}$ | $\lvert T_{cur} - P_{cue}\rvert$ |
| $PE_{int}$ | Prediction error of $P_{int}$ | $\lvert T_{cur} - P_{int}\rvert$ |
| $PE_{prev}$ | Task switch effect | $\lvert T_{cur} - T_{prev}\rvert$ |
| $PE_{joint}$ | Prediction error $P_{joint}$ | $\lvert T_{cur} - P_{joint}\rvert$ |
| $C_{cue}$ | Modulation of $PE_{cue}$ on RT | |
| $C_{int}$ | Modulation of $PE_{int}$ on RT | |
| $\beta$ | scaled reliance on the pre-cue | $\frac{C_{cue}}{C_{cue}+C_{int}}$ |
| $\tilde{\beta}$ | Randomly sampled reliance on the pre-cue | |
| | Proactive switch demand | $\lvert T_{prev} - P_{joint}\rvert$ |
| | Confidence of joint task prediction | $\lvert P_{joint} - 0.5\rvert$ |
| | Proactive interference effect | $\lvert P_{int}-P_{cue}\rvert$ |

DOI: https://doi.org/10.7554/eLife.39497.005

To link these variables to RTs, the unsigned prediction error (discrepancy between the task-set weighting and actual task, denoted as PE) was calculated for each trial and each variable (i.e., $PE_{prev}$, $PE_{cue}$, and $PE_{int}$; *Figure 3B*), where a larger PE indicates a greater need to adjust one's task-set to the actual task demand. Based on the observation that trials with larger PE of the forthcoming task have slower RTs (*Waskom et al., 2017*), we assumed that RTs scale positively with PE. The two rival hypotheses were then modeled as general linear models (GLMs) consisting of the variables defined above. In the max-benefit model, $PE_{cue}$ was included to represent the effect of external cues. To maximize the utility of the informative pre-cue, the $PE_{int}$ that represented the non-informative trial history was not included in this model. Finally, $PE_{prev}$ was used to account for the classic task switch effect. The joint-guidance model consisted of $PE_{cue}$ and $PE_{int}$ to represent the modulations of cue-induced and trial history-based task predictions, respectively. The classic task switch effect in this model was accounted for by $PE_{int}$, because the most recent task has already been factored into $P_{int}$. Our model comparison explored the full model space by also including an alternative model with all three variables ($PE_{prev}$, $PE_{int}$, and $PE_{cue}$), in order to capture any additional effect from the previous trial (e.g., task-set inertia) above and beyond its contribution to $P_{int}$. We also included three control GLMs representing each of the three variables by themselves. Note that the model with only $PE_{prev}$ represents the classic task switch effect. All models also included a constant regressor, in order to model the portion of RT data that do not vary as a function of the present experimental manipulations. To account for mean RT differences in color and motion trials, RTs from the two tasks were fit separately.

Model comparison was conducted using cross-validation to prevent over-fitting and to control for different numbers of free parameters used in the candidate models (*Chiu et al., 2017*) (see Materials and methods: Modeling and model comparison). The performance of the different models was then submitted to Bayesian model comparison (*Stephan et al., 2009*), which calculated protected model exceedance probabilities (i.e., the likelihood of a given model providing the best explanation of the behavioral data) for each candidate model. The joint-guidance model clearly outperformed all other

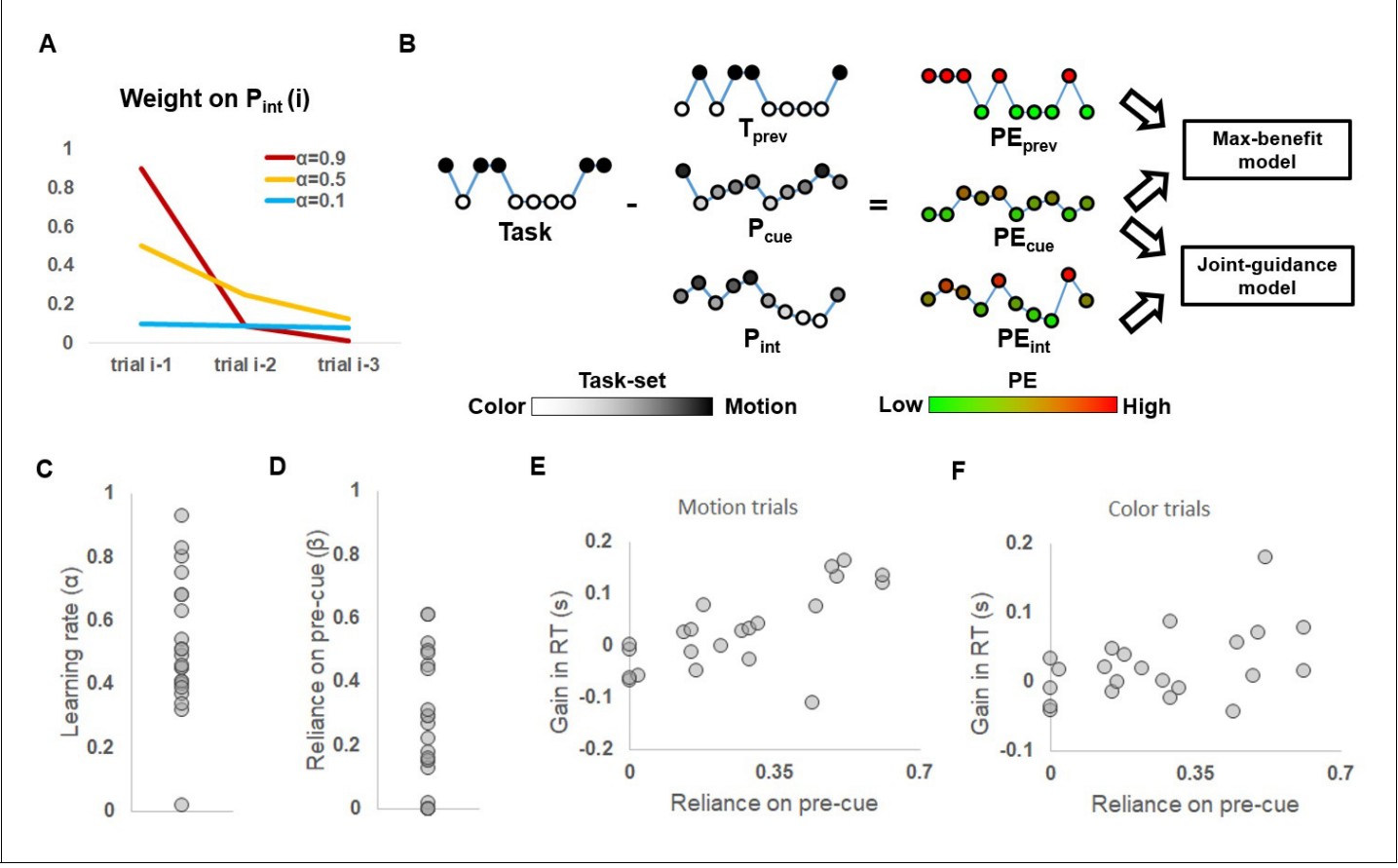

**Figure 3.** Model-based behavioral results. (A) Weights of older trials on determining $P_{int}$, plotted as a function of different learning rates. (B) Illustration of model comparison. Given a trial sequence of tasks and the time course of a model variable (depicted as a string of nodes, with the height and brightness of the nodes coding the task-set), its corresponding time course of (unsigned) PE (with the height and the color of the nodes coding the magnitude of the PE) was calculated. The max-benefit model consisted of $PE_{prev}$ and $PE_{cue}$; and the joint-guidance model consisted of $PE_{int}$ and $PE_{cue}$. (C) Distribution of individual learning rates. (D) Distribution of the reliance on pre-cue relative to the self-generated task prediction. (E, F) Gain in RT when the pre-cue was informative relative to when the pre-cue was non-informative, plotted as a function of the reliance on pre-cue. (E) and (F) show results for motion and color trials, respectively.

DOI: https://doi.org/10.7554/eLife.39497.006

models, with a protected exceedance probability of 0.997, indicating that behavior was best explained by a collective contribution to proactive task-set updating from cue-induced and internally generated task predictions. The joint-guidance model was hence used for all subsequent behavioral and neuroimaging analyses.

## Behavioral data – Quantifying respective contributions of cue-based and trial history-based task predictions

We next sought to more closely characterize how participants combined internally generated and external contributions to task predictions. We began by asking to what degree individual participants relied on an extended trial-history in generating a task prediction, as captured by the RL model's learning rate. The learning rates estimated from individual participants displayed substantial inter-subject variance (range: 0.02–0.93, *Figure 3C*). The mean learning rate was 0.52, indicating that, on average, participants weighted the i-1 trial about as much as the prior trial history in determining the internally generated task prediction.

To quantify participants' relative weighting of trial history-based versus explicit cue-based predictions, we calculated the scaled reliance on the pre-cue (denoted as β, range: 0 to 1) for each participant by $\frac{C_{cue}}{C_{cue}+C_{int}}$, where $C_{cue}$ and $C_{int}$ are the coefficients of $PE_{cue}$ and $PE_{int}$, respectively, after fitting

the joint-guidance model to RTs. Thus, a higher β indicates stronger reliance on the pre-cue and hence weaker dependence on internally generated task prediction; a β of 0.5 indicates equal reliance on $P_{cue}$ and $P_{int}$. Strikingly, we found that even though trial history was not predictive of the forthcoming task, its effect on behavior was on average 3 times as strong as that of the cue-induced task prediction (group mean β: 0.26; range: 0-0.61; one sample t-test against 0.5: $t_{21}$ = 5.42, P < 0.001; *Figure 3C*). Notably, five participants showed either no (β = 0, four subjects) or very little (β = 0.02, one subject) reliance on cue-induced task prediction. Even after excluding these participants (to rule out the possibility that the differential reliance on trial history was due to a failure to understand the associations between the pictorial pre-cue and the task-set prediction), the mean β of the remaining 17 participants remained significantly lower than 0.5 ($t_{16}$=3.96, P = 0.001), again indicating stronger reliance on the internally generated task prediction. To test the robustness of the β estimates, we computed β separately using the first and last 3 runs. Across subjects, β estimates were significantly correlated between these two task phases (r = 0.63, p = 0.002), suggesting reliable β estimation within participants. Also, there was no significant difference in β estimates between the first and last 3 runs ($t_{21}$=1.32, p = 0.20), suggesting that the reliance on $P_{cue}$ relative to $P_{int}$ remained unchanged throughout the 9 runs in the fMRI session.

We have cast the joint-guidance approach to control predictions as sub-optimal, due to the fact that trial-history was not predictive of task transitions. To corroborate this assumption, we quantitatively assessed whether relying more on the internally generated, trial-history based predictions than on explicit, cue-induced task predictions incurs a performance cost. We first estimated the acceleration in responding due to utilizing the pre-cue for each participant and each task. Specifically, the 0.5 (uninformative) prediction level condition was used as a baseline. Then, for each pre-cue/task (motion task vs. color task) combination, we calculated the respective probabilistic expectation of acceleration in RT relative to the 0.5 prediction level (i.e., the probability of encountering this pre-cue/task combination x the RT difference between this combination and the baseline). Across participants, the mean estimated gain of RT was positively correlated with β estimates in both the motion (r = 0.74, p<0.001; *Figure 3E*) and color (r = 0.46, p=0.03; *Figure 3F*) tasks, indicating a clear behavioral benefit for relying on the external cue. This analysis underlines the sub-optimal nature of relying on internally generated, trial history-based task prediction in the present context. We speculate that this seemingly irrational reliance on internally generated task prediction may be attributable to a relatively lower cost (e.g., due to high automaticity) of using internally generated control predictions compared to using cue-based predictions (cf. *Shenhav et al., 2013*; *Shenhav et al., 2017*; see Discussion).

In sum, the behavioral and modeling results clearly demonstrate that task demand predictions were jointly informed by internally generated and externally provided information. Moreover, in spite of being sub-optimal in terms of potential performance benefits, task performance depended more on trial history-based task predictions than on the explicit informative pre-cues. To characterize the brain mechanisms by which internally generated and cue-induced task predictions guide cognitive control, we next turned to interrogating the concurrently acquired fMRI data.

## fMRI data – analysis strategy

The joint guidance model holds that cognitive control is guided both by internally generated and externally cued task prediction:

$$P_{joint} = (1 - \beta)P_{int} + \beta P_{cue} \tag{2}$$

We here sought to characterize how this joint influence is instantiated at the neural level. The initial major question we sought to answer was whether cue-based and history-based expectations influence control in parallel or whether these predictions are in fact integrated in a single brain region. Moreover, we sought to characterize two additional key computations that are required for translating $P_{joint}$ into successful task-set updating (*Figure 4A*): first, after the onset of the pre-cue, the task-set needs to be proactively shifted from $T_{prev}$ to $P_{joint}$ in anticipation of the predicted task demand. The demand for preparatory task-set updating (proactive switch demand) can thus be quantified as |$T_{prev}$ - $P_{joint}$|. Second, following the presentation of the actual task cue and stimulus, the task-set weighting (if not perfectly corresponding with the cued task) needs to be updated *reactively* from $P_{joint}$ to the actual task demand. The reactive switch demand can thus be quantified as

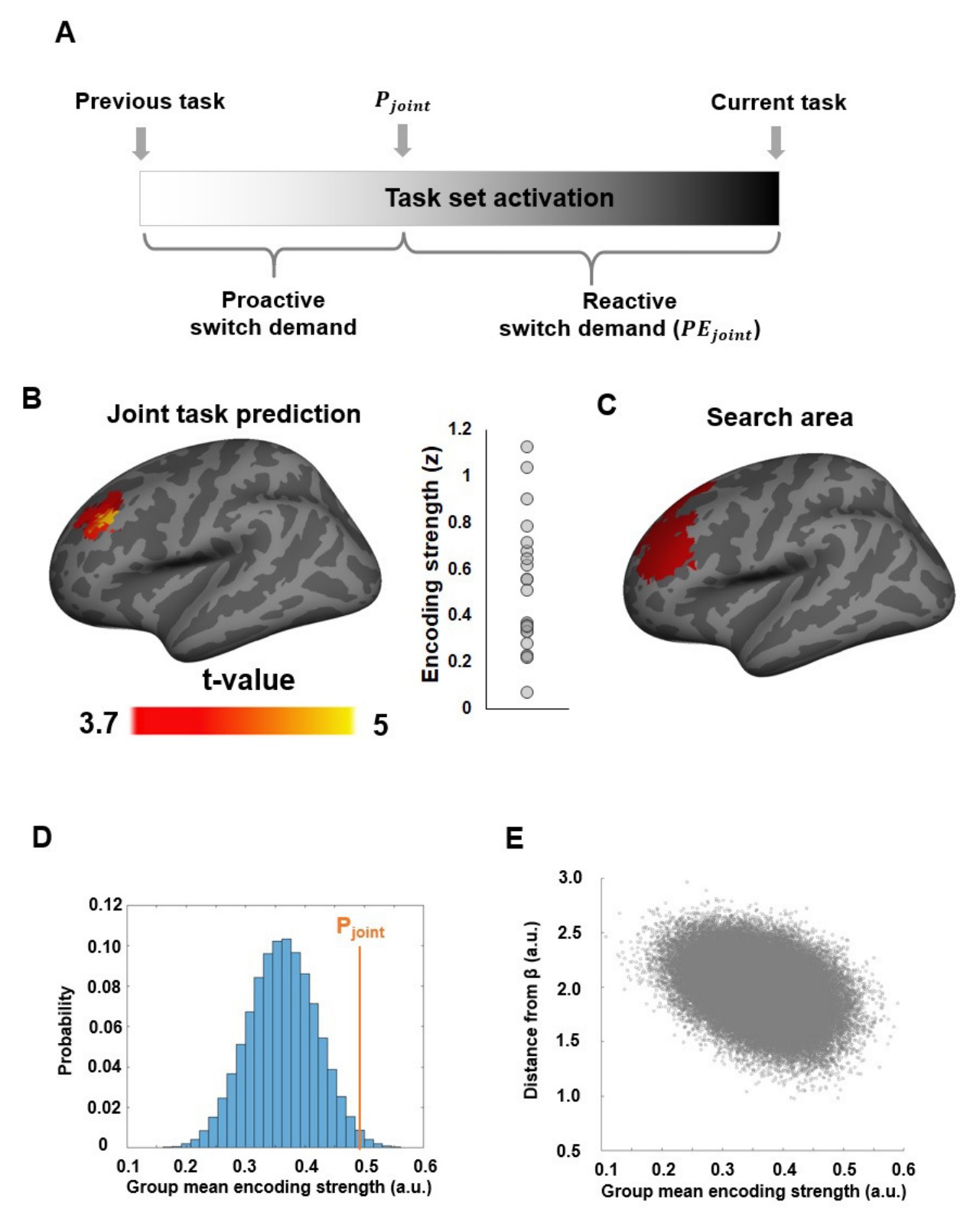

**Figure 4.** Neural representation of the joint task prediction. (A) Illustration of how $P_{joint}$ is translated into proactive and reactive switch demand in relation to previous and forthcoming task demand. (B) Left: An MFG region showing significantly above-chance encoding of the joint task prediction. Right: Individual ROI-mean encoding strength (in z-score). (C) The dlPFC ROI (in red) defined by any linear combination of $P_{cue}$ and $P_{int}$ representation. *Figure 4 continued on next page*

*Figure 4 continued*

(D) Histogram showing the encoding strength of pseudo-$P_{joint}$ based on randomly sampled β parameters using fMRI data in (B). (E) Encoding strength shown in (D), plotted as a function of the distance from $P_{joint}$.

DOI: https://doi.org/10.7554/eLife.39497.007

The following figure supplements are available for figure 4:

**Figure supplement 1.** To validate the MVPA approach, we first leveraged the classic task switching effect observed in the behavioral data, and tested whether task transitions (task-repeat trials vs. task-switch trials) could be decoded from fMRI data time-locked to the onset of the task cue and stimulus, as shown in previous studies (*Jimura et al., 2014*; *Qiao et al., 2017*).

DOI: https://doi.org/10.7554/eLife.39497.008

**Figure supplement 2.** In addition to the expected task-set, $P_{joint}$ can also be thought of as indicating the confidence of the prediction, with the lowest degree of confidence reflected by $P_{joint}$ = 0.5 (i.e., no preference for either task).

DOI: https://doi.org/10.7554/eLife.39497.009

the prediction error of $P_{joint}$, or $PE_{joint}$. Hence, we conducted fMRI analyses to locate brain regions carrying significant information about trial-by-trial variations in these three key variables: the joint task prediction ($P_{joint}$), the proactive switch demand ($|T_{prev} - P_{joint}|$), and the reactive switch demand ($PE_{joint}$).

Given that the motion and color task-sets contain multiple dimensions of task information (e.g., whether the goal was to identify motion direction or color, which color/motion direction was mapped onto which key, frame color, etc.), their neural representations may differ with respect to both mean activity levels and multivariate activity patterns in local voxel clusters. Therefore, in examining the neural representations of the key variables above, we employed searchlight multivoxel pattern analysis (MVPA; see Methods: MVPA procedure) that relies on both multi-voxel activity patterns and univariate activity levels. As a validation, we replicated the classic task-switch effect using this approach (*Figure 4—figure supplement 1A*).

## fMRI data – Encoding of joint task predictions at pre-cue onset

We started by probing for a possible integrated neural representation of the joint externally and internally guided task prediction. Because $P_{joint}$ is a weighted sum of $P_{int}$ and $P_{cue}$, it is inherently correlated with both variables. To ensure that we identify regions that are specifically representing the integrated prediction only, we filtered out searchlights that showed significant encoding of either cue-induced or internally generated predictions (*Figure 4—figure supplement 1B–D*). This analysis produced a map of the spatial distribution of the representation strength of $P_{joint}$, exclusively revealing a left dorsolateral prefrontal cortex (dlPFC) region centered on the middle frontal gyrus (MFG) (*Figure 4B*). To rule out the possibility that the MVPA encoding of $P_{joint}$ in dlPFC merely reflected a univariate task effect (e.g., the color task evoking stronger mean activity due to being more difficult than the motion task), we performed a univariate control analysis. Specifically, within each fold of 3 runs, the searchlight-means of trial-wise t-maps were correlated with the corresponding trial-wise $P_{joint}$ estimates to obtain the z-score of their linear correlation. The mean z-scores averaged across the 3 folds were then used as the estimate of the univariate encoding strength of $P_{joint}$. This approach ensured maximal similarity with the MVPA. Importantly, this whole-brain univariate analysis did not reveal any regional encoding of $P_{joint}$ after correcting for multiple comparisons (voxel-wise threshold $p<0.001$ and cluster size >62 searchlights). Moreover, an ROI-based analysis focusing on the dlPFC region shown in *Figure 4B* showed that the mean univariate encoding strength of $P_{joint}$ did not significantly differ from 0 (z-score = $-0.006 \pm 0.03$, one sample t-test: $t_{21}$ = $-0.20$, p=0.85). Thus, the dlPFC results were not driven by univariate effects of task difficulty.

By definition, $P_{joint}$ is also inherently correlated with any linear combination of $P_{int}$ and $P_{cue}$. Therefore, the left dlPFC region identified above may in principle encode a different mixture of $P_{int}$ and $P_{cue}$ than $P_{joint}$. To rule out this possibility, we conducted a permutation test using randomly sampled βs for each subject (denoted as $\tilde{\beta}$). To ensure this analysis was not biased towards obtaining selective $P_{joint}$ effects, it was based on a search space in a left dlPFC area generated from an F-statistical map that measured the effect caused by any linear combination of $P_{cue}$ and $P_{int}$. The search space was then defined as left dlPFC searchlights with uncorrected P value less than 0.01 in the resulting F-map (244 searchlights in total, *Figure 4C*). Then, $\tilde{\beta}$ was randomly sampled for each participant and

was applied to the same MVPA procedure, in order to gauge the encoding strength of its corresponding pseudo-$P_{joint}$. Group-level analysis was then conducted to determine the largest cluster size that showed significant ($P < 0.001$ uncorrected) encoding of the pseudo-$P_{joint}$. This procedure was repeated 100,000 times, resulting in a null distribution of the largest cluster size. The mean of this null distribution was significantly smaller than the largest cluster size obtained using behaviorally derived βs ($P = 0.02$), suggesting that $P_{joint}$ better accounted for dlPFC fMRI activity patterns than other mixtures of $P_{int}$ and $P_{cue}$.

As a post-hoc analysis, we further examined the selectivity of $P_{joint}$ encoding, relative to other randomly sampled $\tilde{\beta}$ s within the dlPFC cluster shown in *Figure 4B*. To this end, we compared the encoding strength of $P_{joint}$ to the encoding strength of arbitrarily mixed values of $P_{cue}$ and $P_{int}$. Similar to the previous analysis, MVPA was conducted to measure the encoding strength of the pseudo-$P_{joint}$. This procedure was repeated 100,000 times in order to ensure a robust estimation of the underlying null distribution. The group mean encoding strength (i.e., how well the fMRI data in the searchlights fit the model variable in a cross-validation procedure) of $P_{joint}$ was significantly stronger than the chance level derived from this random sampling procedure ($P = 0.02$, *Figure 4D*).

Moreover, we examined whether the behaviorally derived βs represent a local maximum state in the space of all possible βs. A maximum state implies that, given a set of $\tilde{\beta}$, the encoding strength of its corresponding pseudo-$P_{joint}$ should steadily decay as $\tilde{\beta}$ becomes more distinct from β (measured by the Euclidean distance across subjects in the present analysis). On the contrary, if there exist other maxima, this decay would not be present, because distant $\tilde{\beta}s$ would also achieve high encoding strength if they lie close to other maxima. Supporting the idea of β representing a local maximum, we observed a significant negative correlation between $\tilde{\beta}s'$ distance from β and its encoding strength of how well the fMRI data fit the pseudo-$P_{joint}$ ($r = -0.37$, $P < 0.001$; *Figure 4F*). Thus, these results offer strong evidence for a specific role of this dlPFC region in integrating joint predictions of forthcoming task demand, and against the alternative possibility that explicit externally and (likely implicit) internally generated predictions might drive task-set updating independently, without being integrated. We also probed for the neural encoding of the relative strength, or confidence, of task prediction (inverse of prediction uncertainty, $|P_{joint} - 0.5|$), which was represented in frontal and parietal regions (*Figure 4—figure supplement 2*).

## fMRI data – Encoding of proactive switch demand at pre-cue onset

While $P_{joint}$ provides predictions about the forthcoming task, the degree to which accommodating this prediction requires proactive task-set updating depends on its relationship to the previous trial ($|T_{prev} - P_{joint}|$). We here searched for brain regions that encoded this distance between the expected and prior task-set, and thus, the relative need to engage in preparatory task-set updating, or proactive switch demand. Regions encoding this demand are likely responsible for the actual reconfiguration of the task-set. We found encoding of proactive switch demand to be supported by a wide network of regions centered in frontal and parietal cortex (*Figure 5A*). Areas encoding switch demand included right frontopolar cortex (FPC, BA 10), left inferior frontal gyrus (IFG, pars opercularis), left precentral gyrus, bilateral supplementary motor area (SMA), left inferior parietal lobule/intraparietal sulcus (IPL/IPS), bilateral superior parietal lobule (SPL), precuneus, bilateral insula and the bilateral putamen of the dorsal striatum. This network extended well into visual cortex, including bilateral lingual gyrus and bilateral middle occipital gyrus, suggesting that participants also used task predictions to adjust visual processing of forthcoming information.

A core set of these regions (lateral PFC, SMA, and lateral posterior parietal cortex, *Figure 5B*) are responsive to multiple and changing task demands (*Cole et al., 2013*; *Ruge et al., 2013*), are functionally connected to each other (*Yeo et al., 2011*), and have been conceptualized as a frontoparietal cognitive control network (e.g., *Duncan, 2013*), while the dorsal striatum has long been proposed to contribute to updating of working memory content (*Frank et al., 2001*). A subset of these regions was also found to represent the confidence of task predictions (*Figure 4—figure supplement 2*). The current results suggest that the frontoparietal control network is not only involved in exerting control during task execution but also in the anticipatory updating of task-set representations driven by joint internally and externally generated predictions about forthcoming tasks.

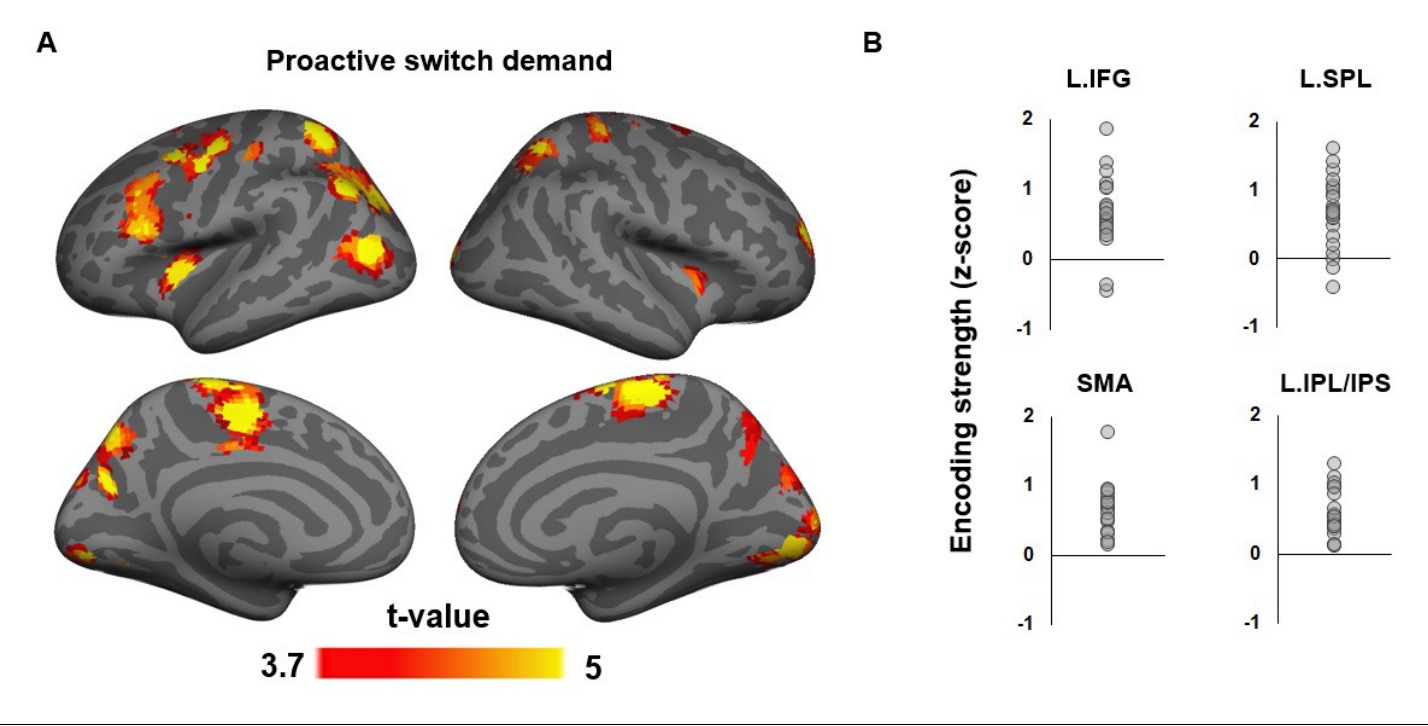

**Figure 5.** Neural encoding of proactive task switch demand. (A) T-statistics maps of brain regions showing significantly above-chance encoding of proactive task switch demand at the onset of the pre-cue. (B) Individual ROI-mean encoding strength of proactive task switch demand.
DOI: https://doi.org/10.7554/eLife.39497.010

## fMRI data – proactive interference

An alternative interpretation regarding model variable $P_{int}$ (and associated findings) is that they may reflect, or be confounded by, proactive interference from previously activated task-sets. In the current modeling framework, at the time of pre-cue onset the degree of proactive interference that would be exerted from prior trials' task-sets can be quantified as the discrepancy between the internal and external predictions, or $|P_{int}-P_{cue}|$. Using this metric to test whether there are any regions that encoded proactive interference, we performed the equivalent analyses on $|P_{int}-P_{cue}|$ that we had previously performed for $P_{int}$ and $P_{cue}$. This failed to reveal any above-chance encoding of $|P_{int}-P_{cue}|$ in whole-brain MVPA, which suggests that proactive interference is unlikely to have played a major role in contributing to our data. In fact, this observation seems to fit with prior studies of proactive interference effects. For instance, using computational modeling and behavioral data, *Badre and Wagner, 2006* showed that proactive interference effects decreased exponentially as a function of cue-task stimulus interval over a time range from 250 ms to 1150 ms. As the intervals in the current study were substantially longer (ranging from 1750 ms to 2500 ms), we speculate that any proactive interference effects may have decayed too much to be a major contributor in modulating proactive task switch preparations in this protocol.

## fMRI data – encoding reactive switch demand at task stimulus onset

We next sought to characterize the neural substrates of reactive switch demand, that is, the need for additional task-set reweighting once the task cue and stimulus are presented, as represented by the prediction error of the joint task prediction ($PE_{joint}$). To control for the influence of the actual task demand, $PE_{joint}$ encoding strength was computed separately for motion and color trials, and the statistical analysis was then performed on data collapsed across the two tasks. This analysis revealed encoding of $PE_{joint}$ in a set of regions consisting of left dmPFC (including the ACC), bilateral precentral and postcentral gyrus, precuneus, right SPL, left inferior occipital gyrus, and IFG (*Figure 6A*).

We also tested the encoding strength of $P_{joint}$ on both color and motion trials at the time of task-stimulus onset and did not find any brain areas passing the correction for multiple comparisons. In

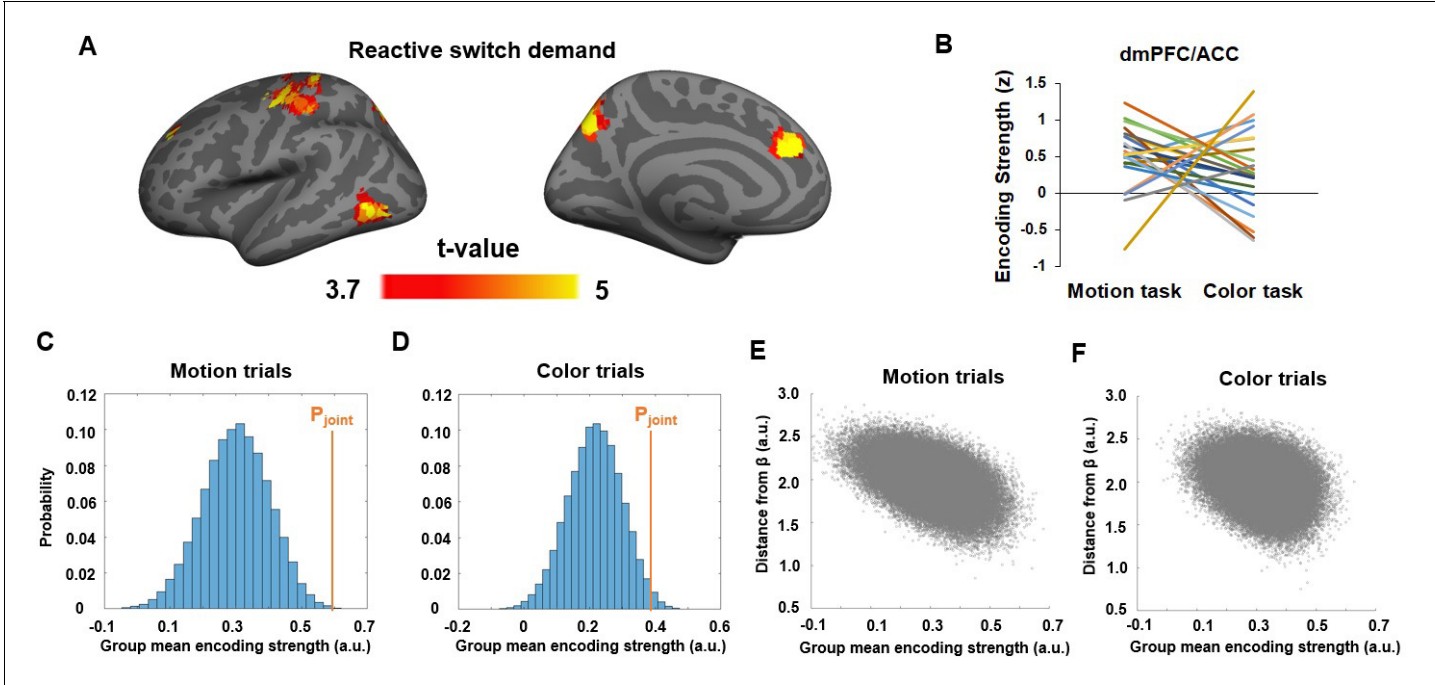

**Figure 6.** Neural representation of the joint task prediction error (PE$_{joint}$). (**A**) T-statistics maps of brain regions showing significantly above-chance encoding of PE$_{joint}$. (**B**) Encoding strength (z-score) of PE$_{joint}$ in the dmPFC/ACC cluster, plotted as a function of task. Each line represents one subject. (**C,D**) Histogram showing the encoding strength of pseudo-PE$_{joint}$ based on randomly sampled β parameters and fMRI data in the dmPFC/ACC cluster in motion (**C**) and color (**D**) trials. (**E**) Encoding strength shown in (**C**), plotted as a function of the distance from PE$_{joint}$. (**F**) Encoding strength shown in (**D**), plotted as a function of the distance from PE$_{joint}$.

DOI: https://doi.org/10.7554/eLife.39497.011

The following figure supplement is available for figure 6:

**Figure supplement 1.** MVPA results showing neural representation of prediction error of internal task prediction at the onset of the task stimulus.
DOI: https://doi.org/10.7554/eLife.39497.012

conjunction with the encoding of P$_{joint}$ at the onset of the pre-cue, this result corroborates the expectation that the representation of P$_{joint}$ and PE$_{joint}$ should be temporally separated. We next compared the degree to which PE$_{joint}$ was selectively represented based on behaviorally derived βs among the regions showing strong encoding of PE$_{joint}$. Compared to randomly sampled βs, the dmPFC (*Figure 6B*), right precentral and postcentral gyrus, and precuneus showed significantly above-chance encoding strength for PE$_{joint}$, with the dmPFC exhibiting the strongest effect. The effects remained above chance in the dmPFC when tested separately using motion (P < 0.001; *Figure 6C*) and color trials (P = 0.03; *Figure 6D*). Furthermore, for a given set of $\tilde{\beta}$s, the encoding strength of its corresponding pseudo-PE$_{joint}$ decreased as a function of its Euclidean distance from βs for both motion (r = -0.51, P < 0.001; *Figure 6E*) and color trials (r = -0.29, P < 0.001; *Figure 6F*). Thus, we obtained strong evidence for an involvement of the dmPFC/ACC in the reactive updating of task demand predictions based on the joint-guidance of internally generated and externally provided cue information.

Finally, we sought to relate the current data set to findings from a recent study that traced neural substrates of (unsigned) PE of internally generated task predictions in a similar task-switching paradigm, but where internal predictions were driven by varying the likelihood of each task being cued over blocks of trials (*Waskom et al., 2017*). The corresponding analysis in the current data set is to search for regions that encode PE$_{int}$ at task-stimulus onset. In close correspondence to the results of *Waskom et al., 2017*, we observed robust encoding of PE$_{int}$ in the frontoparietal control network and the adjacent parietal portion of the dorsal attention network (*Figure 6—figure supplement 1*) (*Yeo et al., 2011*). These data show that the updating of internally generated task predictions derived from a non-predictive trial sequence (in the current study) recruits the same neural

substrates as updating of task predictions in response to predictive trial sequences (*Waskom et al., 2017*).

## Discussion

The human brain is capable of anticipating forthcoming task demand and exerting proactive control to reconfigure information processing in line with those predictions (*Shenhav et al., 2013*; *Egner, 2014*; *Jiang et al., 2014*; *Abrahamse et al., 2016*; *Waskom et al., 2017*). In the present study, we sought to characterize how this regulation of anticipatory control is implemented in the context of concurrent external, cue-based, and internally generated, cognitive history-based predictions of forthcoming control demand in the form of task-set updating. By directly manipulating and formally modeling cue-based and cognitive history-based task predictions, we demonstrated that behavior was driven by joint predictions from external and internal sources, and that these predictions were integrated in (left) dlPFC prior to task stimulus onset. The discrepancy between the joint prediction and the previous-trial task, which signals the demand of proactively shifting task-set, was represented in frontoparietal regions belonging to the frontoparietal control or multiple-demand network, along with the insula, putamen and visual areas. Moreover, upon task stimulus presentation, reactive updating of task-set based on the joint task-demand prediction error was found to be encoded most prominently in the dmPFC.

We conducted a rigorous test of the reliance on internally generated predictions by making them non-informative. Despite of the lack of validity in predicting the forthcoming task, trial history-based, internally generated task prediction exhibited a strong modulation on RTs. This modulation was non-trivial, being about 3 times as strong as the modulation of the informative cue-induced task prediction at the group level. Notably, the finding that behavioral modulation by previous trials could be traced up to three trials back (*Figure 2E–H*) indicates that the modulation of internally generated prediction extends well beyond the scope of the classic task switch effect, which typically concerns only the immediately preceding trial. Formal model comparison also confirmed that the reliance on internally generated task prediction cannot be explained by the classic task switch effect alone.

While these results are consistent with some prior behavioral studies that documented effects of internal, trial history-based predictions even in the presence of 100% informative external cues (*Alpay et al., 2009*; *Correa et al., 2009*; *Kemper et al., 2016*), they are nevertheless surprising, because the strong reliance on the non-informative internally generated task prediction seems to contradict the premise of using such predictions – that is, to optimize the engagement of cognitive control. However, we argue that these results indicate that the process of applying proactive cognitive control based on concurrent internal and cue-based control demand predictions is itself a result of optimization based on a cost/benefit analysis, as proposed by an influential recent model of control regulation, the expected value of control (EVC) model (*Shenhav et al., 2013*; *Shenhav et al., 2017*). This model considers not only the potential benefits of applying top-down control but also the assumed inherent costs (or effort) of doing so (cf. *Kool et al., 2010*), proposing that the engagement of cognitive control is driven by a cost/benefit analysis that optimizes the predicted gain of applying control relative to its cost. When applied to the conundrum of the apparent overreliance on internal, trial-history based versus, external, cue-based control predictions, the EVC model would predict that this situation could come about if applying internal control predictions incurred a smaller cognitive effort cost than employing external control predictions.

The implied lower costs for internally generated prediction than cue-induced prediction may originate from two, not mutually exclusive sources. First, the cue-induced prediction requires learning and retrieving the associations between an external graphical pre-cue and its corresponding task-set prediction, which may incur a higher cost than internally generated prediction. Second, it is reasonable to assume that internally generated predictions that anticipate the near future to resemble the recent past represent a default cognitive strategy grounded in evolutionary adaptation to an environment of high temporal auto-correlation of sensory inputs (*Dong and Atick, 1995*). Accordingly, it has been shown that ongoing perception (*Fischer and Whitney, 2014*) and decision-making (*Cheadle et al., 2014*) are strongly reliant on recent experience. The present results suggest that the same is true for cognitive control processes: the recent past is (implicitly) employed as a powerful predictor of the immediate future (see *Egner, 2014*; *Jiang et al., 2015b*; *Waskom et al., 2017*). If such self-generated predictions of task demand represent a default mode of control regulation,

their generation is likely cheap and, importantly, it may require substantial cognitive effort to override such predictions, which would be another means of rendering the use of the explicit external cue more costly.

An alternative account could be that the reliance on $P_{int}$ originated from $P_{cue}$. Specifically, if the learning of $P_{cue}$ for each pre-cue is achieved by learning from previous trials sharing the same pre-cue, the 'informativeness 'of the pre-cue specific trial history may render the whole trial history informative, hence promoting the reliance on $P_{int}$. As the learning of $P_{cue}$ asymptotes, $P_{cue}$ would become disentangled from $P_{int}$. This account leads to the key prediction that the reliance on $P_{int}$ (i.e., 1-β) should be higher in the beginning of the task than later on, as the gradual detachment of $P_{cue}$ from trial history would decrease the reliance on trial history over time. However, the lack of difference in β estimates between the first and last 3 runs of the fMRI session does not support this notion. One possibility is that this type of process may have occurred during the stair-casing procedure, and hence did not end up affecting the fMRI session.

The internally generated predictions we observed here took the form of increased expectations of task repetition with increasing run length of a given task (also known as the 'hot-hand fallacy'; *Gilovich et al., 1985*). This contrasts with another intuitive possibility, whereby participants could have predicted a change in task with increasing run length of task repetitions (known as the 'gambler's fallacy'; *Jarvik, 1951*). The fact that we observed the former rather than the latter is in line with our assumption that the internal predictions we measure here are likely implicit in nature, as previous work has demonstrated a dissociation between explicit outcome predictions following the gambler's fallacy and implicit predictions, as expressed in behavior, following the hot-hand fallacy (*Perruchet, 1985*; *Jiménez and Méndez, 2013*; *Perruchet, 2015*). Note also that participants' tendency to follow either fallacy should depend on their beliefs as to whether events are generated randomly (in gambling) or non-randomly (the hot hand). Thus, the fact that internally generated predictions followed the hot-hand pattern further corroborates that participants (likely implicitly) assumed the trial history to be non-random.

The EVC model also predicts that the relative reliance on internally vs. externally generated predictions, which are the outcome of a cost-benefit analysis, will change as a function of the cost and/ or the benefit of engaging the more informative externally generated predictions. Further studies are encouraged to explore other manipulations of benefit and cost (e.g., monetary reward) and test how such manipulations shift the reliance on the informative but costly explicit predictions.

The fMRI data shed light on how the two types of task predictions jointly guide cognitive control in the brain. First, rather than distinct anatomical sources exerting parallel influences on behavior, we detected an integrated representation of task predictions at the onset of the pre-cue in left dlPFC. This is in line with this region's well-established role in representing task rules and strategies (for review, see *Sakai, 2008*; cf. *Waskom et al., 2014*; *Waskom et al., 2017*). Additionally, based on this joint prediction, the anticipated switch demand or amount of required task-set updating was represented in the frontoparietal control network, insula, and dorsal striatum. These regions have long been implicated in task-set regulation, as revealed initially by studies comparing neural activity on task-switch with task-repeat cues or trials (reviewed in *Ruge et al., 2013*), and more recently by studies using multivariate pattern analysis to decode the currently active task-set from frontoparietal cortex (*Woolgar et al., 2011*; *Waskom et al., 2014*; *Loose et al., 2017*; *Qiao et al., 2017*). The present results move beyond these findings by showing that more than just encoding a currently active task rule, the frontoparietal control network is engaged in predicting control demands, in the shape of representing the degree to which a current set has to be updated to suit the forthcoming task (or switch likelihood).

After the onset of the task cue and task stimulus, encoding of the prediction error associated with that joint prediction was found in the dmPFC/ACC, which is consistent with a large literature demonstrating ACC and dmPFC encoding of prediction error in a variety of different contexts (*Holroyd and Coles, 2002*; *Ito et al., 2003*; *Matsumoto et al., 2007*; *Alexander and Brown, 2011*; *Ullsperger et al., 2014*). The representation of joint prediction error also reflects the degree to which the task-set has to be updated reactively; thus, this finding is also consistent with the theory that the ACC monitors performance and signals the need for adjustments in cognitive control (*Botvinick et al., 2001*; *Ridderinkhof et al., 2004*). Finally, we also documented that the neural substrates of prediction error from internally generated predictions in the present study matched those of a recent study where those self-generated predictions were related to probabilistically predictable

task sequences (*Waskom et al., 2017*). This finding indicates that internal predictions derived from a non-predictive trial sequence are encoded in much the same fashion as those derived from a predictable sequence.

Our findings also exhibit a clear temporal segregation between the frontoparietal encoding of the proactive switch demand at the onset of pre-cue (*Figure 5A*) and the dmPFC/ACC encoding of reactive switch demand when the cue and task stimulus were presented (*Figure 6A*). This temporal and functional differentiation maps closely onto the 'dual mechanisms of cognitive control' framework (*Braver, 2012*), which distinguishes between proactive, anticipatory, and reactive, compensatory application of control. The present results strongly suggest that the lateral frontoparietal control network guides proactive control implementation, whereas the dmPFC/ACC detects the prediction error of control demand prediction, and signals the need of (or perhaps implements) reactive control to match the actual (rather than anticipated) cognitive control demand. In support of this contention, in a recent study that explicitly distinguished between proactive and reactive control in a conflict task, a left IFG area overlapping the IFG area in *Figure 5* was also implicated in using proactive control demand predictions (*Jiang et al., 2015a*). Furthermore, disrupting function in this area using transcranial magnetic stimulation selectively diminished the effects of learned control predictions on behavior (*Muhle-Karbe et al., 2018*).

Finally, the current findings suggest a novel extension of the EVC theory, which originally addressed a cost/benefit analysis of engaging control *after* prediction of control demands are formed (*Shenhav et al., 2013*). The present results suggest that a cost/benefit optimization process is also applicable to the preceding stage, where different inputs to the control demand prediction process are reconciled. This demand prediction reconciliation process would not only precede the putative EVC calculation, but should also directly influence it, and do so in a unidirectional, hierarchical fashion. More generally, in light of recent theoretic proposals of the hierarchical architecture of cognitive control (*Koechlin and Summerfield, 2007*; *Badre and Nee, 2018*), an overarching optimization procedure may simultaneously consider the costs and benefits of multiple cognitive control processes across different levels in the hierarchy, in order to guide goal-directed behavior while taking into account both the benefit and the cost of engaging cognitive control.

In conclusion, we combined a probabilistic cued task switching with computational modeling and neuroimaging to show that concurrent externally and internally derived predictions of cognitive control demand are reconciled to form a joint prediction. Behavior was dominated by internal, trial-history based predictions, likely due to the lower cost of generating or higher cost of overriding these predictions. The integrated prediction was encoded in dlPFC and guided proactive cognitive control over task-set updating, which was represented in the frontoparietal control network. Subsequently, if actual control demand deviates from predicted demand, the dmPFC/ACC is engaged to compensate the prediction error of the joint prediction, and guide reactive updating. In this manner, the present findings reveal that flexible human behavior depends on multiple regulatory processes that govern cognitive control.

# Materials and methods

## Subjects

Twenty-eight volunteers gave informed written consent, in accordance with institutional guidelines. All subjects had normal or corrected-to-normal vision. Data from six subjects were excluded from further analysis due to low (<50%) accuracy in at least one of the cells in the experimental design (see below). The final sample consisted of 22 subjects (15 females; 22–35 years old, mean age = 27 years). This study was approved by the Duke University Health System Institutional Review Board.

## Experimental procedures

Visual stimuli were presented on a back projection screen viewed via a mirror attached to the scanner head coil. Tasks and response collection were programmed using Psychtoolbox 3 (*Brainard, 1997*) in Matlab (Mathworks, inc.). All visual stimuli were presented in the center of the screen over a grey background. Each trial started with a presentation of a pie chart (radius ≈ 2.2° of visual angle) for 0.5 s. The relative areas of black vs. white regions indicated the probability of a black vs. white frame surrounding the imperative dot cloud later in the trial (see below). Five probability levels

were used in this study: 20%, 40%, 50%, 60% and 80% (applied to both black and white colors). The probability of seeing a black vs. white frame always summed up to 1, as the black and white area always occupied the whole pie chart. To make the perceptual appearance of the predictive cue different across trials, the pie chart rotated by a random degree on each trial. Following the pie chart, a fixation crosshair was presented for an exponentially jittered duration between 1.75 and 2.5 s (step size = 0.25 s). The fixation crosshair was followed by a cloud of 60 colored (either purple or green, radius ≈ 0.15°) moving (speed randomly drawn from a uniform distribution from 13°/s to 15°/s) dots. The dot cloud spanned approximately 6° of visual angle both vertically and horizontally and lasted for 1.5 s. For each trial, the colors and motion directions of the dots were defined by their respective noise levels (ranging from 0.1 to 0.9, determined by a stair-case procedure described below), a dominant motion direction (left or right) and a dominant color (green or purple). For example, a combination of color noise level of 0.8 and motion noise level of 0.4 means that: (1) 20% (i.e., 1–0.8) of all dots were randomly selected to have the dominant color; (2) the remaining 80% of dots were randomly colored in green or purple with equal probability; (3) 60% (i.e, 1–0.4) of all dots were randomly chosen to move in the dominant direction; (4) the remaining 40% of dots had random motion directions; and (5) the dots with the dominant color were selected independently from the dots with the dominant motion direction.

The dot cloud was surrounded by a frame, whose color (either black or white) was predicted by the preceding pie chart. Depending on the color of the frame, participants had to judge the dominant color (green vs. purple) or motion direction (left vs. right) of the dot cloud via two MR-compatible button boxes (one for each hand). The association between frame color and task and the response mappings (e.g., left hand: middle/index finger = green/purple; and right hand: index/middle finger = left/right) were counterbalanced across subjects. If no response was detected upon the offset of the dot cloud, a warning message ('Please respond faster!') was shown on the screen for 2 s. Finally, a second fixation cross (representing the inter-trial interval) was presented for an exponentially jittered duration between 3.5 and 5 s (step size = 0.5 s). If applicable, the duration of the warning message was deducted from this duration. The main task consisted of 9 runs of 50 trials each. Other factors were counterbalanced in a within-run manner, including (a) that each of the five probabilities appeared in 10 trials and (b) that each frame color, dominant color, and consistent motion direction appeared in 25 trials. Importantly, the sequences of tasks were pseudo-randomly produced so that the tasks performed on previous trials had no predictive power on the task to be performed on the current trial.

Prior to fMRI scanning, participants performed a practice session of 20 trials (ITI = 2 s for all trials) to ensure that they comprehended the task instructions. The practice session was followed by a stair-case procedure (4 runs of 50 trials each) that adaptively and separately adjusted the noise levels for color and motion to achieve accuracy of ~87.5% for both color and motion trials (cf. *Waskom et al., 2017*). The trial structure and counterbalancing were identical to the main task. The noise levels for color and motion both started at 0.5 and were re-evaluated respectively at every 5$^{th}$ color and motion trial (check points) based on two rules: (1) If at most one error was made since the last check point, the noise level for the check point's corresponding target feature increased by 0.025; and (2) if the noise level for a feature did not change at any of the past 4 check points, its corresponding noise level decreased by 0.1.

## Behavioral analysis

Error trials and outlier trials (RTs outside subject mean ±3 SDs) were removed from further analyses. Two repeated measures ANOVAs were conducted on both accuracy and RT data. The first ANOVA concerned the effect of the task at the previous trial (previous task: color or motion ×current task: color or motion). The second ANOVA focused on the effect of predictive cues (task prediction: 20%, 40%, 50%, 60% or 80% × current task: color or motion). Similar repeated measure ANOVAs were also conducted by replacing the previous trial with trials i-2 and i-3, in order to test the modulation of older trials on behavior. Note that a 3-way ANOVA (previous task ×task prediction×current task) was not performed due to low trial counts for unexpected task conditions (e.g., only 9 trials for the condition of a motion trial following a color trial and having wrong prediction of 80% chance of encountering a color trial).

## Modeling and model comparison

Rival models in model comparison were GLMs with a subset of trial-wise estimates of $PE_{prev}$, $PE_{cue}$, and $PE_{int}$. Based on the observation that larger PE slows down responses (*Waskom et al., 2017*), a nonnegative constraint was applied to the coefficients (*Chiu et al., 2017*). Each model also included a constant regressor. To compare models, behavioral data were divided into 3 folds, each of which consisted of data from 3 runs. Two folds were used as training data to fit GLMs to RTs. To account for the main effect of task in RT, fitting was performed separately for trials with color and motion tasks. The resulting fitting coefficients were then applied to the same GLMs to predict trial-wise RTs in the remaining fold (test data). This procedure was repeated until each fold served as test data once. Model performance was measured by the product of trial number and the logged average squared trial-wise PE of RTs from all 3 test folds and was calculated for each model and each subject.

## MRI acquisition and preprocessing

Images were acquired parallel to the AC–PC line on a 3T GE scanner. Structural images were scanned using a T1-weighted SPGR axial scan sequence (146 slices, slice thickness = 1 mm, TR = 8.124 ms, FoV = 256 mm * 256 mm, in-plane resolution = 1 mm * 1 mm). Functional images were scanned using a T2*-weighted single-shot gradient EPI sequence of 42 contiguous axial slices (slice thickness = 3 mm, TR = 2 s, TE = 28 ms, flip angle = 90°, FoV = 192 mm * 192 mm, in-plane resolution = 3 mm * 3 mm). Preprocessing was done using SPM8 http://www.fil.ion.ucl.ac.uk/spm/. After discarding the first five scans of each run, the remaining images underwent spatial realignment, slice-time correction, and spatial normalization, resulting in normalized functional images in their native resolution. Normalized images were then smoothed using a Gaussian kernel with 5 mm full-width-half-maximum to increase signal (*Xue et al., 2010*). Single trial fMRI activity levels at the onset of the pre-cue were estimated separately following *Mumford et al. (2012)* by regressing the fMRI signals against a GLM consisting of HRF–convolved onsets of the pre-cue at the trial, the onsets of all other pre-cues, and the onsets of all task stimuli. Other regressors of no-interest, such as the estimated motion parameters, mean white matter (WM) BOLD signal, mean cerebrospinal fluid BOLD signal were also included in the GLM. Single trial fMRI activity levels at the onset of the task stimulus were calculated similarly. For each trial, the resulting t-maps were subtracted by their respective mean in the WM mask, in order to reduce non-neural noise in t estimates across individual t-maps, such as the noise introduced by the different GLMs with partially overlapping regressors of varying co-linearity in the estimation of trial-level t-maps. The adjusted t-maps were then used in MVPA.

## MVPA procedures

We conducted searchlight-based (r = 2 voxels) MVPAs to quantify the representation strength of a given variable (e.g., $P_{cue}$). MVPAs were conducted on a grey matter (GM) mask that was generated by dilating GM voxels (GM value >0.01) in the segmented T1 template by 1 voxel (*Jiang et al., 2015a*). For each searchlight, data from the 9 runs and the trial-wise variable time course were chronologically divided into 3 folds (3 runs per fold), based on which a 3-fold cross-validation was performed. For the training folds, trial-wise fMRI activity levels from all masked GM voxels within the searchlight were fit to the variable time course, resulting in one weight for each voxel. The fitting took the form of a ridge regression (*Xue et al., 2010*) to control for over-fitting. These weights were then applied to the fMRI data in the test fold to produce a predicted variable time course (*Figure 7*). High linear correlation between predicted and actual variable time courses indicated that the variable was represented in the neural data. No correlation (i.e., a correlation coefficient of 0) would indicate no representation of the variable of interest. Since each fold was used as test data once, three correlation coefficients were obtained. We Fisher-transformed these 3 correlation coefficients and used their mean as a quantification of representation strength for the searchlight. After searchlight analyses were performed across the whole brain (using each GM voxel as searchlight center once), a representation strength map was generated, with the center voxel of each searchlight encoding the degree to which each searchlight represents the variable in question.

For group-level analyses, a one-sample t-test against 0 was then conducted on each searchlight's center voxel of the individual representational strength maps using AFNI's 3dttest++ program. To correct for multiple comparisons, group-level results were corrected using a non-parametric

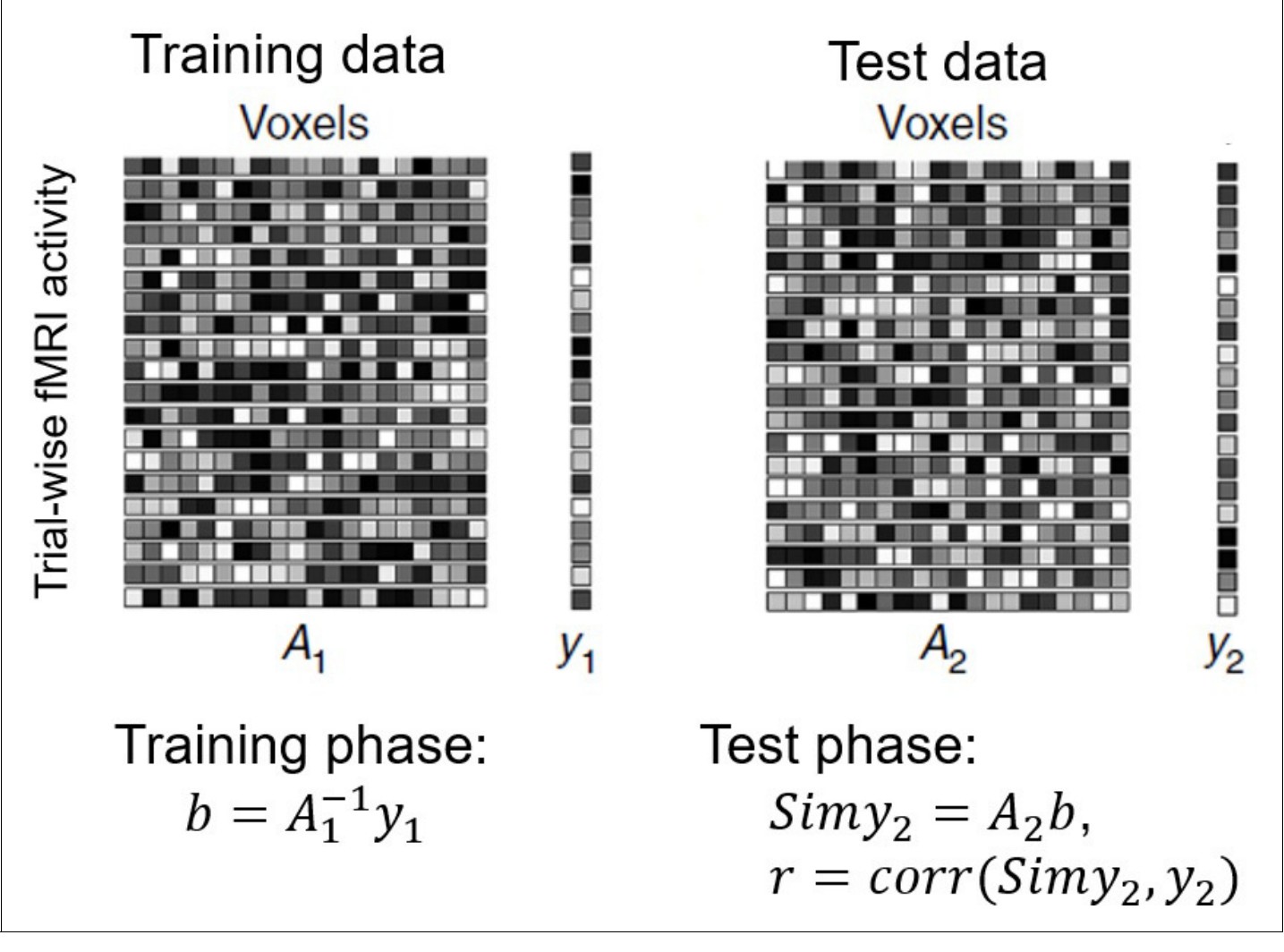

**Figure 7.** Illustration of the MVPA procedure. Training data were used to estimate the weights that fit (either linear-fitting or logistic regression) voxel-wise estimated fMRI activity ($A_1$) to model estimates ($y_1$) in a trial-by-trial manner. The weights were then applied to test fMRI data ($A_2$) to predict the hypothetical BOLD signal in test data ($y_2$). The goodness of prediction, or presentation strength, is measured by the correlation between predicted BOLD signal ($Simy_2$) and $y_2$.
DOI: https://doi.org/10.7554/eLife.39497.013

approach (https://afni.nimh.nih.gov/pub/dist/doc/program_help/3dClustSim.html) that randomly permutated the data for 10,000 times, in order to generate a robust null distribution of the statistical map. When voxelwise threshold was set at $p<0.001$ and family-wise error rate was set at 0.05, this approach yielded cluster size thresholds ranging from 16 to 23 searchlights, depending on the specific analysis.

## Acknowledgements
We thank Anthony Sali for assistance with data collection and Michael L Waskom for helpful comments on a previous version of this manuscript. This project was supported in part by National Institute on Aging awards F32 AG056080 (JJ) and R21 AG058111 (ADW) and National Institute of Mental Health award R01 MH097965 (TE).

# Additional information

## Funding

| Funder | Grant reference number | Author |
|---|---|---|
| National Institute on Aging | F32 AG056080 | Jiefeng Jiang |
| National Institute on Aging | R21 AG058111 | Anthony D Wagner |
| National Institute of Mental Health | R01 MH097965 | Tobias Egner |

The funders had no role in study design, data collection and interpretation, or the decision to submit the work for publication.

## Author contributions

Jiefeng Jiang, Conceptualization, Data curation, Formal analysis, Funding acquisition, Validation, Investigation, Visualization, Methodology, Writing—original draft; Anthony D Wagner, Supervision, Investigation, Methodology, Writing—review and editing; Tobias Egner, Conceptualization, Supervision, Funding acquisition, Investigation, Methodology, Project administration, Writing—review and editing

## Author ORCIDs

Jiefeng Jiang ⓘD http://orcid.org/0000-0002-4264-6382
Anthony D Wagner ⓘD http://orcid.org/0000-0003-0624-4543
Tobias Egner ⓘD http://orcid.org/0000-0001-7956-3241

## Ethics

Human subjects: Twenty-eight volunteers gave informed written consent, in accordance with institutional guidelines. This study was approved by the Duke University Health System Institutional Review Board.

## Decision letter and Author response

Decision letter https://doi.org/10.7554/eLife.39497.020
Author response https://doi.org/10.7554/eLife.39497.021

# Additional files

## Supplementary files

• Transparent reporting form
DOI: https://doi.org/10.7554/eLife.39497.014

## Data availability

Statistical maps for all whole-brain fMRI analyses have been uploaded to https://neurovault.org/collections/3732/. Original behavioral and fMRI data are available at https://openneuro.org/datasets/ds001493. The MATLAB source code for the task and key analyses has been made available on GitHub (https://github.com/JiefengJiang/eLife2018; copy archived at https://github.com/elifesciences-publications/eLife2018).

The following datasets were generated:

| Author(s) | Year | Dataset title | Dataset URL | Database, license, and accessibility information |
|---|---|---|---|---|
| Jiefeng Jiang, Anthony D Wagner, Tobias Egner | 2018 | Statistical maps for all whole-brain fMRI analyses | https://neurovault.org/collections/3732/ | Publicly available at the NeuroVault database. (accession no. 3732) |
| Jiefeng Jiang, An- | 2018 | Original behavioral and fMRI data | https://openneuro.org/ | Publicly available at |

thony D Wagner,
Tobias Egner

datasets/ds001493

the OpenNeuro
website

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
