## [Decision Letter]

[Editors’ note: a previous version of this study was rejected after peer review, but the authors submitted for reconsideration. The first decision letter after peer review is shown below.]

Thank you for submitting your work entitled "Integrated External and Internally Generated Task Predictions Jointly Guide Cognitive Control in Prefrontal Cortex" for consideration by *eLife*. Your article has been reviewed by a Senior Editor, a Reviewing Editor, and three reviewers.. The following individuals involved in review of your submission have agreed to reveal their identity: Carlo Reverberi (Reviewer #2).

Our decision has been reached after consultation between the reviewers. Based on these discussions and the individual reviews below, we regret to inform you that this submission will not be considered further for publication in *eLife*.

This paper takes a novel and sophisticated approach to an important problem. Understanding how internal predictions are integrated with external cues in order to manage task sets and control behavior is of high importance. The reviewers and editors all recognized this importance, and were impressed by the sophisticated approach to modeling behavior and the brain.

The "internal prediction" side of this problem is central to the theoretical impact of this study. However, as the reviews and the subsequent discussion of them made evident, it was unclear what this internal prediction reflects in this study. There appear to be several alternatives, with fairly different implications. For example, rather than an internal prediction, the internal effect could reflect some type of proactive interference (PI). Reviewer 2 points out that a more model-based internal prediction might have made qualitatively different predictions than PI that were not tested here. If true, this would complicate the interpretation of P_joint_, particularly if it did make different predictions. Reviewer 3 further raised a separate point that learning might occur over joint external/internal prediction, rather than keeping them separate.

After considerable deliberation, we decided that there is substantial work to be done to treat a number of alternative accounts of what is happening here and to specify the nature of internal prediction in this task. Some could be done with additional modeling; others might require additional data collection. At *eLife*, we only invite a resubmission in cases where a single revision is likely to conclusively address any major points. The amount to do here and the uncertain outcome of that process is more substantial than would be typical for this type of revision. So, we have decided to reject the paper in its current form. However, in light of the strengths we note above, if you were to undertake the extensive additional analysis and/or data collection required to better characterize these data, we would be willing to consider the paper at *eLife* as a new submission. Though, we must note, that this invitation does not guarantee that the new submission would be reviewed or accepted.

*Reviewer #1*:

The present study examined how internally and externally cued task sets are integrated using quantitative analysis of behavior and fMRI data. Externally cued task sets consisting of visually present probabilistic cues. Internally cued tasks sets were modeled by how much recent trial history affected present task performance. Each source (internal and external) can be formalized as a prediction of the forthcoming task. Violations of these predictions (i.e. prediction errors; PEs) were hypothesized to reduce performance. The authors found that both internal and external PEs slowed performance (or conversely, that accurate predictions sped performance). Based on behavior, individual reliance on both sources was estimated and related to fMRI signals. Joint task prediction was related to signals in the DLPFC, preparatory task-set updating was related to frontal-parietal cortices, and PEs were related to activation in the dmPFC. Collectively, these data indicate separable contributions of multiple control regions to flexible behavior.

There is a lot to like about this study, particularly regarding the formal quantification of internal and external predictions/PEs. I do feel as though there is some opacity with some of the methodological choices, which may breed misunderstanding. Indeed, it is possible that my most substantive concern stems from such misunderstandings. So, perhaps additional clarification is all that is needed.

Essential revisions:

If I understand correctly, P_joint_ reflects the prediction of the color task, which was the harder of the two tasks. So, if activation correlated positively with P_joint_, that could reflect preparation for the color task or preparation for cognitive load (e.g., much like a univariate analysis of a color-word Stroop task would look if contrasting color vs word). It's quite difficult to discern among these possibilities and more general task-set prediction with the MVPA procedure employed by the authors. As I understand it, the MVPA procedure is very similar to a traditional univariate analysis, at least initially, with the key differences being that (A) a neighborhood of voxels is regressed onto a predictor rather than a single voxel, and (B) ridge regression is used to induce regularization. Given that the data are smoothed, I suspect that the use of a searchlight rather than a single voxel simply induces more smoothing (i.e. effectively mimic'd by a larger smoothing kernel during preprocessing), so then the only real difference is the use of ridge regression vs OLS. Then, where the methods differ substantively is that inference from a univariate method would interpret the sign of the resulting β values (e.g., positive is activation), whereas the sign is effectively removed in the MVPA procedure in the cross-validation step. In this case, cross-validation can succeed if the betas are positive in both the training and test data (i.e. more activity in the DLPFC in preparation for the color task/harder task), negative in both the training and test data (i.e. less activity in the DLPFC in preparation for the color task/harder task), or a mixture of the two. The mixture would be what would indicate an abstract task prediction. However, given the 30 or so years we've been contrasting harder tasks vs easier tasks and seeing the DLPFC engaged more for the former, I'm worried that what is being depicted is preparation for the harder task rather than an abstract task set. There is still something very interesting about that prediction being formulated by both internal and external sources, but the interpretation is quite different (e.g., representation vs processing).

Note that if the authors had used MVPA to predict the task itself (i.e. classification) this would not be an issue. Given the association between MVPA and classification, I'm worried that many readers will make this mistake. I think that this can be investigated fairly easily by either (A) doing a simple univariate analysis using P_joint_ as a parametric modulator, or (B) examine the βs produced by the ridge regression procedure. If the DLPFC region is positive on these metrics, then it would seem it largely reflects preparation for the harder task.

Even if this holds true, all is not lost! I believe the data depicted in Figure 4—figure supplement 2 is what we really want here. Those data indicate the absolute deviation from chance (i.e. deviation from no prediction). I think that's really what a task set would reflect. So, perhaps it is as simple as swapping in Figure 4—figure supplement 2 for Figure 4 and changing the story from the DLPFC to rostrolateral PFC and the IPS.

*Reviewer #2*:

The manuscript contrasts two types of anticipatory control (i.e., control based on a prediction of future states of the environment): one "external" based on explicit cues, another "internal" based on observed history.

Notwithstanding no information on the next task is present in the task history, subjects seem to rely on it, even more than on informative cues.

The manuscript is interesting and well written. Nevertheless, I am not fully convinced on the interpretation that the authors offer on one of the primary measures.

If I correctly understood Figure 3 and the description of the model, P_int_ for the color task increases monotonically with the length of the most recent color-task series. Thus, to be clear, between these three series:a) […] c-m-c-c-m-m-mb) […] c-m-c-c-m-m-cc) […] m-m-c-m-c-c-c

P_int_-color is c>b>a

In other words, subjects would strongly expect another color task in c) while they would be highly surprised to get a color task in a).

The task sequence, however, is actually (pseudo)random:

- There is no dependency between history and next trial

- The proportion of the two tasks is 50/50

- (I guess that) the distribution of the length of same-task series is roughly exponential with a mode of 1 and a median of 2 (?).

Given that, the subjects may rather fast realize that the probability of a long same-task series is a priori very low compared to a short one. The usual (invalid) psychological reaction to this situation (see e.g. studies on random sequences perception/prediction) is to expect/predict alternation.

Thus, overall, I would expect that for subjects the prediction of the probability of color task would not monotonically increase with the length of the color sequence. For example, I would guess thati) […] cii) […] c ciii) […] c c civ) […] c c c c csubjects would explicitly predict that another c would be more likely in i) or ii) rather than in iii) or iv).

Notice that here I am assuming that sequences like iv or iii were rare in the task. Things would change if this were not the case.

In this experiment, no explicit measures of future task prediction have been collected from subjects, so we cannot know for sure what subjects’ expectations were.

Overall, given the way P_int_ is computed, I would instead consider it a measure of the strength of proactive interference from past trials. This would be consistent with the observation that the interference is stronger with the most extended same-task series.

This view would produce a significant shift in the interpretation of the results.

On another point:

What incentive did the subjects have to perform any control adaptation in advance? Given the task, a "rational" option available to a subject would be to wait for the task phase in which all information is available. For example, in Wisniewski et al., 2015 we used monetary incentives + adaptive timing to keep subjects motivated to use advance information.

*Reviewer #3*:

In the present study by Jiang et al., differing progenitors of task-related demands on cognitive control are assessed behaviorally and neurally. Specifically, explicit demands guided by external cues and implicit demands driven by internal history are contrasted for their predictive impact on cognitive control. A probabilistic task-switching paradigm was utilized to dissociate these two sources of information. Additionally, prediction-error (PE) variables and a Q-learning-inspired computational model were used to more acutely probe the behavioral and neural data. Behaviorally, the findings include support of a 'joint guidance' hypothesis, which states that cognitive control is jointly informed by external and internal information. Neurally, prediction error derived from joint guidance was found to be integrated in a dorsolateral prefrontal cortex (dlPFC) region. Lastly, the demands of proactive task switching and reactive task updating were found to be encoded in the frontoparietal network (FPN) and dorsomedial prefrontal cortex (dmPFC), respectively. The foremost merit of this study is in addressing the varying sources of information that impact cognitive control in a manner that reveals them as distinct neurally. Thus, cognitive control is proposed to contain multiple processes.

Essential revisions:

1) One key assumption in this study is that the task allows external and internal sources of prediction to be separable. Moreover, the task is designed for trial history to be controlled for, such that this history is uninformative (e.g., "set to zero.", Introduction). However, given that there were pre-scan training trials, and many trials included in the scanning (test) sessions ("9 runs at 50 trials each", Discussion section), learning effects might be present that make external and internal information less dissociable. That is to say, even though the trials are randomized, trial history is informative because the probabilistic value of each cue is being stored in a way that constitutes 'internal history'. Therefore, even though it is noted that this task makes trial-history (internal info) and cueing (external info) independent and is exclusively informative on cueing, an alternative perspective is that the task is biased for internal history. After the initial training trials, cue-based predictions may be entangled with internal history, given that probabilities associated with each cue have been learned (approximately). Behavioral and computationally modeled findings reported here might support the latter. Firstly, trial history assessed by i-1 (and i-2 to i-3) trial reaction time biased behavior (subsection “Behavioral data – 130 Effects of external cues and cognitive history”). Secondly, trial history has a three-fold larger effect on behavior than cueing (subsection “Behavioral data – Model comparison.”). Lastly, prediction error for internal history was encoded by networks overlapping with a previous study that had "predictive trial sequences" (Results section). One suggestion is to examine if (and how) the variables derived from these assumptions change over time. More specifically, if such variables differ from run to run. For example, does the internal history prediction variable (P_int_ and related PE_int_) increase or decrease from the first 50 trials to the last 50 trials?

2) Relatedly, given that the intention was to set internal history to zero in terms of predictability, calling this source of information (during modeling and analyses) a type of 'prediction' is a bit confusing. That is to say, if internal history is truly set to zero, how can it be used as an independent factor in analyses and termed a source of prediction?

3) At various points, further context and/or justification for analytic choices would benefit the claims made herein.

3a. Firstly, what justifies using the following combinations that comprise the two guidance models: (1) prediction error related to the previous trial (PE_prev_) and PE of cueing (PE_cue_) amount to the "max benefit hypothesis", and (2) P_cue_ and prediction error related to internal trial-history (PE_int_) amount to "joint guidance" (subsection “Behavioral data – Model comparison”). Even though control models and a model with all three factors were also compared, what is the conceptual backing and/or prior literature supporting this choice?

3b. Next, why is reaction time used for the bulk of the modeling work (and development of variables) as opposed to accuracy?

3c. Why is it assumed that 'optimal performance' is equal to fastest performance? In interpreting results in terms of cost/benefit analyses (e.g., in the expected value of control framework), it is presumed herein that speedy responses are equivalent to optimal responses (subsection “Behavioral data – Quantifying respective contributions of cue-based and trial history-based task predictions.”, and Discussion section). Reaction time gain was found to be positively correlated with using cue-based information, therefore the stronger impact of history-based information on behavior was surprising (subsection “Behavioral data – Quantifying respective contributions of cue-based and trial history-based task predictions.”, and Discussion section), and this was explained in terms of history having a lower cost because it is more automatic. However, it is not clear why internal history would be more automatic and how accuracy factors into this line of reasoning on performance benefits. Mean accuracy was quite high (table 1, Materials and methods section), thus it is possible that the cost to reaction time is outweighed by the benefit to accuracy, and that the observed trial-history-bias supports this.

[Editors’ note: what now follows is the decision letter after the authors submitted for further consideration.]

Thank you for resubmitting your work entitled "Integrated External and Internally Generated Task Predictions Jointly Guide Cognitive Control in Prefrontal Cortex" for further consideration at *eLife*. Your revised article has been favorably evaluated by Michael Frank (Senior Editor), David Badre (Reviewing Editor), and three reviewers.

The reviewers feel that their major concerns were addressed by your revision. They and the editors are in agreement that the manuscript is now acceptable for publication at *eLife*. However, you will note that two of the reviewers raised some additional suggestions of ways the manuscript could be clarified. Though these are not essential revisions, we return this to you once more to give you the opportunity to make changes based on these suggestions and questions. In particular, we encourage you to carefully consider the suggestions that would help make the task paradigm and analysis logic clearer (the first comment from both reviewers). The comments for further clarification from the are copied below.

*Reviewer #2*:

1) Introduction: This was brought up in the first round of review, and adequately addressed in the added analyses. However, concerns over the language used here bears repeating as it was a conceptual obstacle in following the logic of the present paradigm. 'Trial history' is a broad concept that likely has components, and the single component of "trial sequence" is fixed at zero with randomization. This appears to be the point of the chosen paradigm: e.g., that the potential confound of "sequence" is controlled for by randomization, so any internally driven predictions are based on the subject's choice to do so (sub-optimally, as is probed subsection “Behavioral data – Quantifying respective contributions of cue-based and trial history-based task predictions”). That is to say, internally driven factors become independently "discoverable" (via computational modeling) after randomization. The "fixed to zero" description confuses this. Perhaps re-wording this sentence to be less all-encompassing (e.g., it currently reads that all of "trial history", as a singular concept, is fixed at zero) would allay potential confusion on part of the average reader.

1a) Note that the utility of the paradigm became clearer as I read the Materials methods section and Results section, but only became very clear once reading the text. It should be clarified before results are even reported, hence the suggestion to adjust (or otherwise qualify) the phrasing of "fixed at zero".

1b) Note that this also has implications for adjusting the commentary introducing the "rational/max-benefit hypothesis"in subsection “Behavioral data – Effects of external cues and cognitive history”. If a reader doesn't understand that randomization is a beneficial manipulation that allows for computational modeling of the internal factor, then it becomes confusing to suggest that on one hand this paradigm allows us to adjudicate between the impact of external and internal factors (or their joint influence/a comparison), but on the other hand, the internal factor (in its entirety, as is currently suggested) is set to zero, thus the max-benefit hypothesis amounts to explicitly externally-driven processing. The typical reader might wonder: How could the internal factor be part of the adjudication if it's set to zero? Conversely, if there is some discoverable aspect of the internal factor, how could it be discounted from the rational hypothesis (in principle, not mathematically)?

*Reviewer #3*:

The reply of the authors clarified my concerns. I have only a few further comments.

It is now clearer what the authors meant for internal prediction. The use of the word "prediction" both for internal and external prediction misguided me to think that both predictions would be explicit, i.e., the subject would be aware of the prediction. The authors argue that this is not the case: while the cue-based prediction is explicit, the internal prediction is implicit and likely unconscious. Even more: the authors suggest the possibility of a dissociation between an explicit internal prediction vs. an implicit internal prediction.

For the sake of clarity, the authors may emphasize the qualitative distinction between the two types of predictions also in Introduction and Materials and methods section. Otherwise, the reader might realize it only in discussion. This detail seems important for a correct interpretation of the findings and the paradigm.

Besides, hot hand vs. gambler fallacy both depart from a correct interpretation of chance but the violations go in opposing directions. The major driver of the difference between the two is the belief of the subject about the situation. If she is assuming that the generation of events is random (as in the casino) then she will fall for the gambler fallacy, if she assumes that the generation is not random (as for a basket player) then she may fall in the hot hand fallacy. What do the subjects believe in your task? You did not provide any information on the way sequences are generated, thus in principle participants may hypothesize any of the two. Given the task context, it is likely that the subjects assume a random generation. Thus, they would show the gambler fallacy if asked for an explicit prediction. The effect does not emerge because subject's behavior is dominated by the implicit effect discussed above. If the authors think so, then the mention of the hot hand may be removed.

The fact that subjects relied more on internal predictions might follow from the fact that there was a low/no incentive in performing the task as fast and accurate as possible. In fact, given that there is no incentive subjects may rationally decide to rely more on the cheaper prediction from the sequence, rather than on the most cognitively expensive prediction from cue. Given that, the balance between the two types of prediction might be specific for this task context and it should be generalized with caution.

---

## [Author Response]

[Editors’ note: the author responses to the first round of peer review follow.]

Reviewer #1:[…] 1) If I understand correctly, P_joint_ reflects the prediction of the color task, which was the harder of the two tasks. So, if activation correlated positively with P_joint_, that could reflect preparation for the color task or preparation for cognitive load (e.g. much like a univariate analysis of a color-word Stroop task would look if contrasting color vs word). It's quite difficult to discern among these possibilities and more general task-set prediction with the MVPA procedure employed by the authors. As I understand it, the MVPA procedure is very similar to a traditional univariate analysis, at least initially, with the key differences being that (A) a neighborhood of voxels is regressed onto a predictor rather than a single voxel, and (B) ridge regression is used to induce regularization. Given that the data are smoothed, I suspect that the use of a searchlight rather than a single voxel simply induces more smoothing (i.e. effectively mimic'd by a larger smoothing kernel during preprocessing), so then the only real difference is the use of ridge regression vs OLS. Then, where the methods differ substantively is that inference from a univariate method would interpret the sign of the resulting β values (e.g. positive is activation), whereas the sign is effectively removed in the MVPA procedure in the cross-validation step. In this case, cross-validation can succeed if the betas are positive in both the training and test data (i.e. more activity in the DLPFC in preparation for the color task/harder task), negative in both the training and test data (i.e. less activity in the DLPFC in preparation for the color task/harder task), or a mixture of the two. The mixture would be what would indicate an abstract task prediction. However, given the 30 or so years we've been contrasting harder tasks vs easier tasks and seeing the DLPFC engaged more for the former, I'm worried that what is being depicted is preparation for the harder task rather than an abstract task set. There is still something very interesting about that prediction being formulated by both internal and external sources, but the interpretation is quite different (e.g. representation vs processing).Note that if the authors had used MVPA to predict the task itself (i.e. classification) this would not be an issue. Given the association between MVPA and classification, I'm worried that many readers will make this mistake. I think that this can be investigated fairly easily by either (A) doing a simple univariate analysis using P_joint_ as a parametric modulator, or (B) examine the βs produced by the ridge regression procedure. If the DLPFC region is positive on these metrics, then it would seem it largely reflects preparation for the harder task.

We thank the reviewer for raising this important point, which allowed us to rule out a potential alternative interpretation for the dlPFC findings. We followed the reviewer’s first suggestion, i.e., to perform a univariate control analysis, as this approach allows for whole-brain analysis. To ensure that the results of the univariate analyses are maximally comparable to the MVPA results, we kept the identical (MVPA) procedure, with the only change that we used the (univariate) searchlight-mean for each trial-level t-map to form a time course of the t-values, which was then correlated to the time course of P_joint_ within each of the 3 folds. The resulting correlation coefficients were then transformed into z values. The mean z values across the 3 folds were taken to reflect the univariate encoding strength. This approach corresponds to the reviewer’s suggestion in that: (1) similar to the parametric modulator, the correlation between the t-value time course and trial-level P_joint_ captures how brain activity co-varies with model estimates; and (2) the use of the searchlight mean mimicked stronger smoothing while not having the smoothing kernel exceed the searchlight size. Using the same multiple-comparison correction procedure as in the MVPA (voxel-wise threshold P < 0.001 and cluster size > 62 searchlights using AFNI’s 3dttest++), no brain areas showed significant positive or negative correlations between searchlight-mean t-values and P_joint_. When using the dlPFC region shown in Figure 4B as an ROI, the ROI mean univariate encoding strength (z-score = -0.006 ± 0.03) did not significantly differ from 0. This outcome indicates that the contribution of univariate activity to the MVPA results appears to be rather limited. This finding also rules out the possibility that the multivariate encoding of P_joint_ in the dlPFC could be solely attributed to increased univariate activity when the color task was predicted.

These new results are reported in the revised manuscript in the following manner:

“To rule out the possibility that the MVPA encoding of P_joint_ in dlPFC merely reflected a univariate task effect (e.g., the color task evoking stronger mean activity due to being more difficult than the motion task), we performed a univariate control analysis. […] Moreover, an ROI-based analysis focusing on the dlPFC region in Figure 4B showed that the mean univariate encoding strength of P_joint_ did not significantly differ from 0 (z-score = -0.006 ± 0.03, one sample t-test: t_21_ = -0.20, P = 0.85). Thus, the dlPFC results were not driven by univariate effects of task difficulty.” (subsection “fMRI data – Encoding of joint task predictions at pre-cue onset”)

Following reviewer 1’s comment 11, we also performed the same univariate analyses on P_cue_ (Multiple comparisons correction results: voxel-wise threshold: P < 0.001, cluster size > 68 searchlights) and P_int_ (Multiple comparisons correction results: voxel-wise threshold: P < 0.001, cluster size > 64 searchlights). These analyses revealed significant negative correlations between P_cue_ and searchlight-mean t-values in bilateral posterior temporal gyrus (MT+) and medial prefrontal cortex. Critically, these regions do not overlap with any of the MVPA results (Figure 4—figure supplement 1B), suggesting that the MVPA results cannot be accounted for by aggregate univariate effects. These results were integrated into the revised Figure 4—figure supplement 1 and its figure legend.

In sum, by following the reviewer’s suggested analysis approach, we successfully ruled out the possibility that our main dlPFC finding reflects a univariate task difficulty effect. We thank the reviewer for encouraging these additional analyses, as they provide further support for the inferences drawn.

Following the reviewer’s guidance, we have now also uploaded all t-maps to NeuroVault (https://neurovault.org/collections/3732/) for interested readers.

2) Even if this holds true, all is not lost! I believe the data depicted in Figure 4—figure supplement 2 is what we really want here. Those data indicate the absolute deviation from chance (i.e. deviation from no prediction). I think that's really what a task set would reflect. So, perhaps it is as simple as swapping in Figure 4—figure supplement 2 for Figure 4 and changing the story from the DLPFC to rostrolateral PFC and the IPS.

We agree with the reviewer that the deviation from neutral prediction shown in Figure 4—figure supplement 2 provides important information about the proactive task switch. Our interpretation is that it reflects the strength or confidence of the task-set prediction. However, unlike P_joint,_ this deviation does not indicate the direction of the task prediction (i.e., whether the motion or color task is more likely to be required next), so we still think that P_joint_ is the more direct reflection of task-set prediction.

Reviewer #2:[…] Overall, given the way P_int_ is computed, I would instead consider it a measure of the strength of proactive interference from past trials. This would be consistent with the observation that the interference is stronger with the most extended same-task series. This view would produce a significant shift in the interpretation of the results.

The reviewer raises a number of very pertinent points. First, regarding the speculation on subjects’ expectations, our behavioral findings/modeling suggest that, as the reviewer notes, expectations for task A increase with increasing length of runs of task A trials. This runs counter to intuition and also to some studies of explicit predictions, which tend to follow the “gambler’s fallacy” (Jarvik, 1951) that the reviewer describes. However, importantly, this is not the case for implicit predictions, which tend to show the opposite pattern (the “hot-hand fallacy”; Gilovich et al., 1985). This dissociation between explicit and implicit predictions (as measured by behavior) has been demonstrated very convincingly in seminal work by Pierre Perruchet, (1985), and is now known as the Perruchet effect (for a recent review, see Perruchet, 2015). Importantly, this dissociation has also been demonstrated for predictions of task demands in the realm of cognitive control (Jimenez and Mendez, 2013). So, the fact that behavior in our task seems to follow the hot-hand rather than gambler’s fallacy is in fact in line with our conceptualization of internal predictions as being implicit. We now point this out explicitly in the revised manuscript in the following manner:

“The internally generated predictions we observed here took the form of increased expectations of task repetition with increasing run length of a given task (also known as the “hot-hand fallacy”; Gilovich et al., (1985)). This contrasts with another intuitive possibility, whereby participants could have predicted a change in task with increasing run length of task repetitions (known as the “gambler’s fallacy”; Jarvik, (1951)). The fact that we observed the former rather than the latter is in line with our assumption that the internal predictions we measure here are implicit in nature, as previous work has demonstrated a dissociation between explicit outcome predictions following the gambler’s fallacy and implicit predictions, as expressed in behavior, following the hot-hand fallacy (Perruchet, 1985; Jimenez and Mendez, 2013; Perruchet, 2015).” (Discussion section)

Second, the reviewer wonders whether our model variable P_int_ (and associated findings) may be better thought of as reflecting proactive interference from previously activated task-sets. This idea has intuitive appeal, but we believe that P_int_ in fact differs from proactive interference in subtle but important ways: specifically, while the P_int_ variable simply captures the (inferred) readiness for a particular task based on the previous task sequence, a proactive interference measure at the time of pre-cue onset should instead correspond to the difference between P_int_ and the (accurate) prediction supplied by the explicit cue value, because P_int_ only “interferes” with the upcoming task to the extent that it differs from that task. Thus, on conceptual grounds alone, we would argue that we should not shift our interpretation from an internal prediction to a proactive interference account.

However, it is of course nevertheless conceivable that proactive interference contributes to our task and the neural data we report. To formally assess this possibility, we conducted additional fMRI analyses and added a section to the revised manuscript in the following manner:

“fMRI data – proactive interference. An alternative interpretation regarding model variable P_int_ (and associated findings) is that they may reflect, or be confounded by, proactive interference from previously activated task-sets. […] As the intervals in the current study were substantially longer (ranging from 1750ms to 2500ms), we speculate that any proactive interference effects may have decayed too much to be a major contributor in modulating proactive task switch preparations in this protocol.” (Subsection “fMRI data – proactive interference”)

Relatedly, please note that the neural encoding of |P_int_-P_cue_| is not necessarily predicted by the joint-guidance model, because that model computes the joint task prediction as a weighted *sum* of P_int_ and P_cue._ Finally, as with all other t-maps, we have now uploaded the t-map for this control analysis to NeuroVault (https://neurovault.org/collections/3732/) for the interested readers.

On another point:What incentive did the subjects have to perform any control adaptation in advance? Given the task, a "rational" option available to a subject would be to wait for the task phase in which all information is available. For example, in Wisniewski et al., 2015 we used monetary incentives + adaptive timing to keep subjects motivated to use advance information.

To keep participants engaged, we used a calibration procedure to make the task challenging. Moreover, participants with low accuracy (possibly due to low motivation) were excluded from behavioral and fMRI analyses. Finally, the assumption that participants performed advance task-set preparation is of course borne out by our behavioral results, which showed a significant main effect of pre-cue prediction on response time.

The above said, we appreciate the reviewer’s point that motivation is likely to be an important determinant of engaging in proactive task switch preparation. We now allude to this issue in a revised line of the Discussion section:

“Further studies are encouraged to explore other manipulations of benefit and cost (e.g., monetary reward) and test how such manipulations shift the reliance on informative but costly explicit predictions.”

Reviewer #3:[…] 1) One key assumption in this study is that the task allows external and internal sources of prediction to be separable. Moreover, the task is designed for trial history to be controlled for, such that this history is uninformative (e.g., "set to zero.", Introduction). However, given that there were pre-scan training trials, and many trials included in the scanning (test) sessions ("9 runs at 50 trials each", Discussion section), learning effects might be present that make external and internal information less dissociable. That is to say, even though the trials are randomized, trial history is informative because the probabilistic value of each cue is being stored in a way that constitutes 'internal history'. Therefore, even though it is noted that this task makes trial-history (internal info) and cueing (external info) independent and is exclusively informative on cueing, an alternative perspective is that the task is biased for internal history. After the initial training trials, cue-based predictions may be entangled with internal history, given that probabilities associated with each cue have been learned (approximately). Behavioral and computationally modeled findings reported here might support the latter. Firstly, trial history assessed by i-1 (and i-2 to i-3) trial reaction time biased behavior (subsection “Behavioral data – 130 Effects of external cues and cognitive history”). Secondly, trial history has a three-fold larger effect on behavior than cueing (subsection “Behavioral data – Model comparison.”). Lastly, prediction error for internal history was encoded by networks overlapping with a previous study that had "predictive trial sequences" (Results section). One suggestion is to examine if (and how) the variables derived from these assumptions change over time. More specifically, if such variables differ from run to run. For example, does the internal history prediction variable (P_int_ and related PE_int_) increase or decrease from the first 50 trials to the last 50 trials?

The reviewer raises an interesting point, that the reliance on external task prediction (P_cue_) and the reliance on trial history based task prediction (P_int_) may be entangled at the beginning of the experiment. In other words, if the learning of P_cue_ for each pre-cue is achieved by learning from previous trials sharing the same pre-cue, the ‘informativeness ‘of the pre-cue specific trial history may render the whole trial history informative, hence promoting the reliance on P_int_. This account of learning processes in our task leads to the key prediction that the reliance on P_int_ (i.e., 1-β) should be higher in the beginning of the task than in later phases, as the gradual detachment of P_cue_ from trial history should decrease the reliance on trial history over time.

To test this possibility, we followed the reviewer’s suggestion and estimated the βs from the first and last run separately, using the same procedure for model-based behavioral analysis. Specifically, for each of the two runs, a grid search of the learning rate was conducted on the joint-guidance model to obtain the learning rate for P_int_ and its corresponding trial-wise PE_int_ that maximized the joint-guidance model’s ability to explain the variance in trial-wise RTs. After the learning rate was determined, the β estimate can then be obtained as a scaled reliance on P_cue_ in relation to the total reliance on P_cue_ and P_int_. We compared β estimates obtained from the first and last run (50 trials per run) and found no difference (paired t-test, t_21_=0.25, p = 0.8).

One potential caveat to this approach is that the relatively low run-wise trial counts may affect the reliability of β estimates, as we found no within-subject consistency of β when correlating the first and last run β estimates across subjects (r = 0.06, p = 0.80). Therefore, we re-ran this analysis using the first and last 3 runs, and obtained robust within-subject reliability (r = 0.63, p = 0.002). The relevant β estimates still did not differ between the first and last 3 runs of the main task, however (t_21_=1.32, p = 0.20). If the cue-specific learning was the source of the reliance on trial history, another prediction would be that reliance on P_cue_ should be stronger than the reliance on P_int_, especially early on in the task, as the former was the source of the reliance on trial history. However, regardless of whether we use only the first run or the first 3 runs, β estimates at the beginning of the main task were significantly lower than 0.5 (one-sample t-tests: t_21_=6.28 and t_21_=6.04, both Ps < 0.001), indicating stronger reliance on P_int_ than P_cue_.

We speculate that there might be two explanations for the data pattern we observe: (1) the disentanglement of P_cue_ occurred prior to the main task, as the stair-case procedure, which preceded the main task, consisted of 200 trials; (2) The entanglement of P_cue_ and P_int_ through trial history may be too weak to be detected in this task. In either case, these analyses do not support the idea that our results reflect an entanglement of internal and external sources of predictions. Nevertheless, we agree with the reviewer that this type of inter-dependence between external and internal sources of predictions is an important possibility in this type of task design, and we now discuss this issue (and associated analyses) in the revised paper in the following manner:

“To test the robustness of the β estimates, we computed β separately using the first and last 3 runs. Across subjects, β estimates were significantly correlated between these two task phases (r = 0.63, p = 0.002), suggesting reliable β estimation within participants. Also, there was no significant difference in β estimates between the first and last 3 runs (t_21_=1.32, p = 0.20), suggesting that the reliance on P_cue_ relative to P_int_ remained unchanged throughout the 9 runs in the fMRI session.” (subsection “Behavioral data – Quantifying respective contributions of cue-based and trial history-based task predictions”)

“An alternative account could be that the reliance on P_int_ originated from P_cue_. Specifically, if the learning of P_cue_ for each pre-cue is achieved by learning from previous trials sharing the same pre-cue, the ‘informativeness ‘of the pre-cue specific trial history may render the whole trial history informative, hence promoting the reliance on P_int_. As the learning of P_cue_ asymptotes, P_cue_ would become disentangled from P_int_. This account leads to the key prediction that the reliance on P_int_ (i.e., 1-β) should be higher in the beginning of the task than later on, as the gradual detachment of P_cue_ from trial history would decrease the reliance on trial history over time. However, the lack of difference in β estimates between the first and last 3 runs of the fMRI session does not support this notion. One possibility is that this type of process may have occurred during the stair-casing procedure, and hence did not end up affecting the fMRI session.” (Discussion section)

2) Relatedly, given that the intention was to set internal history to zero in terms of predictability, calling this source of information (during modeling and analyses) a type of 'prediction' is a bit confusing. That is to say, if internal history is truly set to zero, how can it be used as an independent factor in analyses and termed a source of prediction?

It is a common finding that people intuit non-existing patterns in random sequences (Huettel et al., 2002), and it is in this sense that we call trial sequence a source of predictions. Therefore, even though objectively trial history was non-predictive, participants could still subjectively use trial history to make (invalid) prediction of forthcoming tasks. Importantly, regardless of the validity of the prediction, applying the prediction to proactively shift task-set was expected to have behavioral consequences, such that if the prediction (coincidentally) matched the forthcoming task, behavioral responses would be faster (cf. Waskom et al., 2017). Our behavioral findings clearly support this notion, as we show that participants’ behavior is in fact strongly affected by the non-predictive task sequence.

To clarify this issue further, we first added the following to the revised manuscript to highlight that other studies have also found that trial history modulates cognitive control even when it provides no additional predictive power:

“Previous behavioral studies showed that the two types of predictions appear to drive control simultaneously. In particular, trial history based predictions impact cognitive control even in cases where these predictions are redundant, as in the presence of 100% valid external cues for selecting the correct control strategy (e.g., (Alpay et al., 2009; Correa et al., 2009; Kemper et al., 2016)).” (Introduction)

Second, we also added a discussion on the nature of these predictions, which also relates to a point raised by reviewer 2 (comment 1):

“The internally generated predictions we observed here took the form of increased expectations of task repetition with increasing run length of a given task (also known as the “hot-hand fallacy”; Gilovich et al. (1985)). This contrasts with another intuitive possibility, whereby participants could have predicted a change in task with increasing run length of task repetitions (known as the “gambler’s fallacy”; Jarvik (1951)). The fact that we observed the former rather than the latter is in line with our assumption that the internal predictions we measure here are implicit in nature, as previous work has demonstrated a dissociation between explicit outcome predictions following the gambler’s fallacy and implicit predictions, as expressed in behavior, following the hot-hand fallacy (Perruchet, 1985; Jimenez and Mendez, 2013; Perruchet, 2015).” (Discussion section)

3) At various points, further context and/or justification for analytic choices would benefit the claims made herein.3a. Firstly, what justifies using the following combinations that comprise the two guidance models: (1) prediction error related to the previous trial (PE_prev_) and PE of cueing (PE_cue_) amount to the "max benefit hypothesis", and (2) P_cue_ and prediction error related to internal trial-history (PE-int) amount to "joint guidance" (subsection “Behavioral data – Model comparison”). Even though control models and a model with all three factors were also compared, what is the conceptual backing and/or prior literature supporting this choice?

We thank the reviewer for drawing attention to the need for further justification of the considered models. We have added the following to clarify the choice of the two models:

“In the max-benefit model, PE_cue_ was included to represent the effect of external cues. To maximize the utility of the informative pre-cue, the PE_int_ that represented the non-informative trial history was not included in this model. Finally, PE_prev_ was used to account for the classic task switch effect. The joint-guidance model consisted of PE_cue_ and PE_int_ to represent the modulations of cue-induced and trial history-based task predictions, respectively. The classic task switch effect in this model was accounted for by PE_int_, because the most recent task has already been factored into P_int_.” (subsection “Behavioral data – Model comparison”)

3b. Next, why is reaction time used for the bulk of the modeling work (and development of variables) as opposed to accuracy?

First, RT data, being continuous in nature, tend to be a more powerful and sensitive measure of processing costs, and the task-switching literature has in fact focused on RT switch costs to a much greater extent than on switch costs in accuracy (for reviews, see (Monsell, 2003; Kiesel et al., 2010; Vandierendonck et al., 2010)). Second, in the ANOVA we observed a significant main effect of pre-cue prediction in RT data but not in accuracy. We speculate that one reason may be that trial-wise accuracy is binary and is thus a coarse measure to map onto a continuous variable like PE. Additionally, for incorrect trials, there may be other factors that caused the error (e.g., a trial with low PE may be incorrect due to mind wandering). Therefore, in the subsequent model-based analysis, we used RTs on correct trials to infer the external and internal predictions of task-set. In the revised manuscript, we added the following:

“To formally compare how well the joint-guidance and max-benefit hypotheses explain the behavioral data, we constructed a quantitative model for each hypothesis and compared the two using trial-wise RTs. Accuracy was not modeled due to its insensitivity to external task prediction induced by the pre-cue (Figure 2C).” (subsection “Behavioral data – Model comparison”)

3c. Why is it assumed that 'optimal performance' is equal to fastest performance? In interpreting results in terms of cost/benefit analyses (e.g., in the expected value of control framework), it is presumed herein that speedy responses are equivalent to optimal responses (subsection “Behavioral data – Quantifying respective contributions of cue-based and trial history-based task predictions.”, and Discussion section). Reaction time gain was found to be positively correlated with using cue-based information, therefore the stronger impact of history-based information on behavior was surprising (subsection “Behavioral data – Quantifying respective contributions of cue-based and trial history-based task predictions.”, and Discussion section), and this was explained in terms of history having a lower cost because it is more automatic. However, it is not clear why internal history would be more automatic and how accuracy factors into this line of reasoning on performance benefits. Mean accuracy was quite high (table 1, Materials and methods section), thus it is possible that the cost to reaction time is outweighed by the benefit to accuracy, and that the observed trial-history-bias supports this.

In the instructions, participants were told to respond as fast and accurately as possible, such that “optimal performance” would correspond to fast, correct trials. We agree with the reviewer that, in principle, accuracy may interact with RT to produce a speed-accuracy trade off, such that accuracy improves as RT increases. However, this was not the case in the current study, because (1) there was no main effect of external prediction on accuracy (Figure 2C) and (2) conditions that produced slower responses were also associated with lower accuracy (Figure 2A and 2E). In other words, better accuracy in conditions with faster RTs supports the notion that faster performance is here equivalent with better performance.

In any event, since the term “automatic” may have stronger connotations than we sought to convey, we have now removed the claim that the internal control predictions may be more automatic than external control predictions, and we have made some other minor tweaks to this aspect of the Discussion section. The related text now reads:

“[…] the EVC model would predict that this situation could come about if applying internal control predictions incurred a smaller cognitive effort cost than employing external control predictions. The implied lower costs for internally generated prediction than cue-induced prediction may originate from two, not mutually exclusive sources. […] If such self-generated predictions of task demand represent a default mode of control regulation, their generation is likely cheap and, importantly, it may require substantial cognitive effort to override such predictions, which would be another means of rendering the use of the explicit external cue more costly.”

[Editors' note: the author responses to the re-review follow.]

Reviewer #2:1) Introduction: This was brought up in the first round of review, and adequately addressed in the added analyses. However, concerns over the language used here bears repeating as it was a conceptual obstacle in following the logic of the present paradigm. 'Trial history' is a broad concept that likely has components, and the single component of "trial sequence" is fixed at zero with randomization. This appears to be the point of the chosen paradigm: e.g., that the potential confound of "sequence" is controlled for by randomization, so any internally driven predictions are based on the subject's choice to do so (sub-optimally, as is probed subsection “Behavioral data – Quantifying respective contributions of cue-based and trial history-based task predictions”). That is to say, internally driven factors become independently "discoverable" (via computational modeling) after randomization. The "fixed to zero" description confuses this. Perhaps re-wording this sentence to be less all-encompassing (e.g., it currently reads that all of "trial history", as a singular concept, is fixed at zero) would allay potential confusion on part of the average reader.1a) Note that the utility of the paradigm became clearer as I read the Materials methods section and Results section, but only became very clear once reading the text. It should be clarified before results are even reported, hence the suggestion to adjust (or otherwise qualify) the phrasing of "fixed at zero".

We apologize for the confusion. We have now removed the phrase “fixed at zero” in order to avoid giving the readers a wrong impression that the prediction is a constant. The revised text now reads:

“While the benefit of the explicit cue was represented by its predictive value, trial history was uninformative about the probability of task switching, as the task sequence was randomized. There was thus no objective benefit to generating trial history-based predictions. However, based on prior studies, we nevertheless anticipated…” (Introduction)

1b) Note that this also has implications for adjusting the commentary introducing the "rational/max-benefit hypothesis" in subsection “Behavioral data – Effects of external cues and cognitive history”. If a reader doesn't understand that randomization is a beneficial manipulation that allows for computational modeling of the internal factor, then it becomes confusing to suggest that on one hand this paradigm allows us to adjudicate between the impact of external and internal factors (or their joint influence/a comparison), but on the other hand, the internal factor (in its entirety, as is currently suggested) is set to zero, thus the max-benefit hypothesis amounts to explicitly externally-driven processing. The typical reader might wonder: How could the internal factor be part of the adjudication if it's set to zero? Conversely, if there is some discoverable aspect of the internal factor, how could it be discounted from the rational hypothesis (in principle, not mathematically)?

Following the reviewer’s suggestion, we revised the relevant text in the following manner:

“Given that trial-history was not informative of the upcoming task (i.e., it had no predictive value), this alternative model in effect corresponds to control being guided exclusively by the external cue, which has predictive value. We here refer to this as the ‘max-benefit hypothesis’.” (Results section)

Reviewer #3:The reply of the authors clarified my concerns. I have only a few further comments.It is now clearer what the authors meant for internal prediction. The use of the word "prediction" both for internal and external prediction misguided me to think that both predictions would be explicit, i.e., the subject would be aware of the prediction. The authors argue that this is not the case: while the cue-based prediction is explicit, the internal prediction is implicit and likely unconscious. Even more: the authors suggest the possibility of a dissociation between an explicit internal prediction vs. an implicit internal prediction.For the sake of clarity, the authors may emphasize the qualitative distinction between the two types of predictions also in Introduction and Materials and methods section. Otherwise, the reader might realize it only in discussion. This detail seems important for a correct interpretation of the findings and the paradigm.

In line with the reviewer’s suggestion, we have now made some minor revisions and additions throughout the paper in order to (a) clarify the likely difference in nature between internal (more likely implicit) and external (explicit) predictions, and (b) that the implicit nature of the internal predictions is an assumption (rather than established fact). Specifically, we made sure to avoid calling the predictions “implicit” and instead called them only “internal” while occasionally adding “(likely implicit)” to stress the point that prior literature would suggest these predictions are probably implicit in nature. We made the following changes:

First, please note that we already raised this distinction in the Introduction:

“Importantly, such expectations about task demands can be driven by *two* sources: explicit predictions provided by external cues (Rogers and Monsell, 1995; Dreisbach et al., 2002; Badre and Wagner, 2006) and internally generated, trial history-based predictions, which are typically implicit (Dreisbach and Haider, 2006; Mayr, 2006; Bugg and Crump, 2012; Egner, 2014; Chiu and Egner, 2017).”

Second, in the subsequent section, we now write:

“However, it is not presently known whether and how the brain reconciles explicit external and (typically implicit) internal predictions.”

Further down in the Introduction, we now write:

“However, based on prior studies, we nevertheless anticipated that participants would form (likely implicit) internal expectations for forthcoming trials based on trial-history (e.g., (Huettel et al., 2002)), and this design ensured that trial-history and cue-based predictions were independent of each other.”

In the Results section, we amended a sentence to state:

“Thus, these results offer strong evidence for a specific role of this dlPFC region in integrating joint predictions of forthcoming task demand, and against the alternative possibility that explicit external and (likely implicit) internal predictions might drive task-set updating independently, without being integrated.”

Finally, in the Materials and methods section, we added the following description:

“Importantly, the sequences of tasks were pseudo-randomly produced so that the tasks performed on previous trials had no predictive power on the task to be performed on the current trial.”

Besides, hot hand vs. gambler fallacy both depart from a correct interpretation of chance but the violations go in opposing directions. The major driver of the difference between the two is the belief of the subject about the situation. If she is assuming that the generation of events is random (as in the casino) then she will fall for the gambler fallacy, if she assumes that the generation is not random (as for a basket player) then she may fall in the hot hand fallacy. What do the subjects believe in your task? You did not provide any information on the way sequences are generated, thus in principle participants may hypothesize any of the two. Given the task context, it is likely that the subjects assume a random generation. Thus, they would show the gambler fallacy if asked for an explicit prediction. The effect does not emerge because subject's behavior is dominated by the implicit effect discussed above. If the authors think so, then the mention of the hot hand may be removed.

We thank the reviewer for pointing this out. We believe that reference to the gambler’s and hot-hand fallacy may aid readers who are familiar with these concepts in appreciating an important aspect of our results. We therefore opted to keep mention of the hot hand in the Discussion section. However, we now also add the reviewer’s astute point that the type of fallacy a participant may be subjects to will likely depend on their belief about the randomness underlying the events in question.

Accordingly, we have added the following to the Discussion section:

“The fact that we observed the former rather than the latter is in line with our assumption that the internal predictions we measure here are likely implicit in nature, as previous work has demonstrated a dissociation between explicit outcome predictions following the gambler’s fallacy and implicit predictions, as expressed in behavior, following the hot-hand fallacy (Perruchet, 1985; Jimenez and Mendez, 2013; Perruchet, 2015). Note also that participants’ tendency to follow either fallacy should depend on their beliefs as to whether events are generated randomly (in gambling) or non-randomly (the hot hand). Thus, the fact that internally generated predictions followed the hot-hand pattern further corroborates that participants (likely implicitly) assumed the trial history to be non-random.”

The fact that subjects relied more on internal predictions might follow from the fact that there was a low/no incentive in performing the task as fast and accurate as possible. In fact, given that there is no incentive subjects may rationally decide to rely more on the cheaper prediction from the sequence, rather than on the most cognitively expensive prediction from cue. Given that, the balance between the two types of prediction might be specific for this task context and it should be generalized with caution.

We agree with the reviewer that the balance between internally and externally generated predictions may change depending on other factors such as motivation. In the revised manuscript, we added the following Discussion section:

“The EVC model also predicts that the relative reliance on internally vs. externally generated predictions, which are the outcome of a cost-benefit analysis, will change as a function of the cost and/or the benefit of engaging the more informative externally generated predictions. Further studies are encouraged to explore other manipulations of benefit and cost (e.g., monetary reward) and test how such manipulations shift the reliance on the informative but costly explicit predictions.”